# NSD2 overexpression drives clustered chromatin and transcriptional changes in a subset of insulated domains

Priscillia Lhoumaud[1], Sana Badri[1,7], Javier Rodriguez-Hernaez[1,7], Theodore Sakellaropoulos[1,2], Gunjan Sethia[1], Andreas Kloetgen[1], MacIntosh Cornwell[1], Sourya Bhattacharyya [3,4], Ferhat Ay [3,4], Richard Bonneau [5], Aristotelis Tsirigos [1,6] & Jane A. Skok [1,2]*

CTCF and cohesin play a key role in organizing chromatin into topologically associating domain (TAD) structures. Disruption of a single CTCF binding site is sufficient to change chromosomal interactions leading to alterations in chromatin modifications and gene regulation. However, the extent to which alterations in chromatin modifications can disrupt 3D chromosome organization leading to transcriptional changes is unknown. In multiple myeloma, a 4;14 translocation induces overexpression of the histone methyltransferase, NSD2, resulting in expansion of H3K36me2 and shrinkage of antagonistic H3K27me3 domains. Using isogenic cell lines producing high and low levels of NSD2, here we find oncogene activation is linked to alterations in H3K27ac and CTCF within H3K36me2 enriched chromatin. A logistic regression model reveals that differentially expressed genes are significantly enriched within the same insulated domain as altered H3K27ac and CTCF peaks. These results identify a bidirectional relationship between 2D chromatin and 3D genome organization in gene regulation.

[1] Department of Pathology, New York University Langone Health, New York, NY 10016, USA. [2] Laura and Isaac Perlmutter Cancer Center, NYU School of Medicine, New York, NY 10016, USA. [3] Division of Vaccine Discovery, La Jolla Institute for Immunology, La Jolla, CA 92037, USA. [4] School of Medicine, University of California, San Diego, La Jolla, CA 92093, USA. [5] Department of Biology, Center for Genomics and Systems Biology, NYU, New York, NY 10003, USA. [6] Applied Bioinformatics Laboratories, NYU School of Medicine, New York, NY 10016, USA. [7] These authors contributed equally: Sana Badri, Javier Rodriguez-Hernaez. *email: Jane.Skok@nyulangone.org

The 3D organization of chromosomes enables cells to balance the biophysical constraints of the crowded nucleus with the functional dynamics of gene regulation. Chromosomes are divided into large domains that physically separate into two nuclear compartments, A and B, of active and inactive chromatin, respectively[1]. Loci within each compartment interact more frequently with other loci from the same compartment irrespective of whether these regions are proximal on the linear chromosome. Compartments can be further subdivided into Mbp-sized 'topologically associating domains' (TADs), which are separated from each other by insulating boundaries enriched for the architectural proteins, CTCF and cohesin[2–4]. Since TADs are largely conserved between cell types and across different species they are considered the basic organizational unit of eukaryotic genomes. They play an important role in gene expression by limiting the influence of enhancers to genes located within the same TAD[5–7].

CTCF and cohesin are major contributors in shaping architecture within TADs and depletion of either factor leads to loss of TAD structure[8–10]. Depletion of cohesin also leads to strengthening of compartments, suggesting that compartments and TADs are formed by two different mechanisms and that TAD structures antagonize compartmentalization by mixing regions of disparate chromatin status[9,10]. The best-accepted model to explain TAD formation and maintenance, involves a loop-extrusion mechanism whereby cohesin rings create loops by actively extruding DNA until the complex finds two CTCF-binding sites in convergent orientation[11]. In this way cohesin forms chromatin loops as a result of its ability to hold together two double-strand DNA helices via its ring structure[11–15]. Indeed, genome-wide analyses revealed that loops are preferentially formed between convergently orientated CTCF-binding sites[16] and divergent sites delineate boundary regions[17]. These sites are thought to serve as a block to the movement of cohesin on chromatin.

A number of labs have demonstrated that disruption of a single CTCF-binding site is sufficient to alter chromosomal interactions leading to the spreading of active chromatin and altered gene regulation[14,18–20]. However, it is not known if the reverse is true, and whether alterations in chromatin modifications themselves can impact chromosome organization at the level of A and B compartments, TAD structure, CTCF binding and enhancer–promoter contacts in a manner that corresponds to changes in gene regulation. An alteration in the balance of the antagonistic marks H3K36me2 and H3K27me3 is a hallmark of many different cancers. Translocation between the immunoglobulin heavy chain locus, *IGH* on chromosome 14 with the *NSD2* locus (also known as MMSET or WHSC1) on chromosome 4, leads to NSD2 overexpression in 15–20% of multiple myeloma (MM) patients that have a poor survival rate and do not respond well to cytotoxic chemotherapy[21–24]. NSD2 is a histone methyl transferase that is responsible for deposition of the H3K36 mono-methyl and di-methyl mark. In a wild-type setting, H3K36me2 accumulates on active gene bodies and acts as a signature of transcriptional activity. However, when NSD2 is overexpressed as a result of the 4;14 translocation, H3K36me2 spreads outside of active gene bodies into intergenic regions. Expansion of H3K36me2 domains results in contraction of H3K27me3 domains, altering gene expression programs in the absence of driver mutations in a manner that is poorly understood[25].

Of note, similar changes in chromatin are detected in other cancers such as B and T acute lymphoblastic leukemia (B- and T-ALL) and a number of advanced stage solid tumors, including prostate, colon, and skin cancers[26]. Increased H3K36me2 in some cases can result from an E1099K mutation in NSD2 that affects the catalytic domain of this enzyme[27]. In two pediatric brain cancers, diffuse intrinsic pontine glioblastoma (DIPG) and

supratentorial glioblastoma multiforme (GBMs), a mutation in H3.3 in which the lysine at position 27 is mutated to a methionine (H3K27M) results in a similar H3K36me2 versus H3K27me3 imbalance by impacting the action of EZH2[28–30]. Given the poor prognosis of patients suffering from these cancers, it is important to better understand the mechanisms underlying changes in gene expression in diseases with an H3K36me2 versus H3K27me3 imbalance.

In MM, alterations in gene expression are dependent on the histone methyl-transferase activity of NSD2[31]. Although the impact of NSD2 overexpression on chromatin modifications has been well documented, there is no in-depth analysis into the mechanisms underlying the changes in gene expression that occur downstream of the expansion and reduction of active H3K36me2 and repressive H3K27me3 domains. Using isogenically matched MM patient-derived cell lines that differ only in the levels of NSD2 they express, we demonstrate that spreading of H3K36me2 from active gene bodies into intergenic regions is accompanied by changes in H3K27ac (a feature of regulatory elements) and CTCF binding. Both changes are linked to significant alterations in gene expression and oncogene activation. Expansion of H3K36me2 domains also drives compartment switching and alterations in intra-TAD interactions, while altered boundary insulation scores overlap differential CTCF and Rad21 (a component of the cohesin complex) binding. A logistic regression model reveals that differentially expressed genes are significantly enriched within the same insulated domain as altered H3K27ac and CTCF peaks. These results reveal a bidirectional relationship between 2D and 3D chromatin organization in gene regulation and demonstrate that cells can co-opt altered chromatin domains to drive oncogenic transcriptional programs within insulated boundaries.

## Results

**NSD2 overexpression leads to alterations in H3K27ac.** NSD2 overexpression leads to spreading of H3K36me2 from active gene bodies into intergenic regions leading to a more open chromatin conformation[23]. Alterations in gene expression are dependent on the histone methyl-transferase activity of NSD2, highlighting the importance of deposition and expansion of H3K36me2 domains in the activation of oncogenic transcriptional pathways[31]. However, it is not known how intergenic spreading of H3K36me2 alters gene regulation and whether 3D organization of the genome plays a role.

In order to address this question, we used two previously characterized isogenic cell lines generated from the patient-derived KMS11 t(4;14) MM cell line: non-translocated knockout (NTKO) and translocated knockout (TKO) cells, which have the translocated allele or the endogenous *NSD2* allele, respectively inactivated by insertion of a stop codon just after exon 6, leading to a truncated protein that lacks all functional domains (Fig. 1a)[24]. Importantly, NTKO and TKO cells differ genetically solely in NSD2 expression levels and henceforth are referred to as NSD2 High and Low cells. The paired cell lines allow us to analyze the impact of NSD2 overexpression independent of other confounding genetic alterations found in patient-based samples that do not have appropriate controls. Using RNA-seq we observed that NSD2 High expression leads to the deregulation of many genes (1650 up and 303 downregulated genes) (Fig. 1b, Supplementary Data 1) and principal component analysis (PCA) revealed that NSD2 High and Low replicates separated into distinct clusters (Supplementary Fig. 1a)[23,25], consistent with previous studies[23,25]. MM and KRAS pathways were enriched (FDR < 0.25) as shown by GSEA based on the fold change of expression in NSD2 High vs. Low cells (Supplementary Fig. 1d) (see the "Methods" section for details).

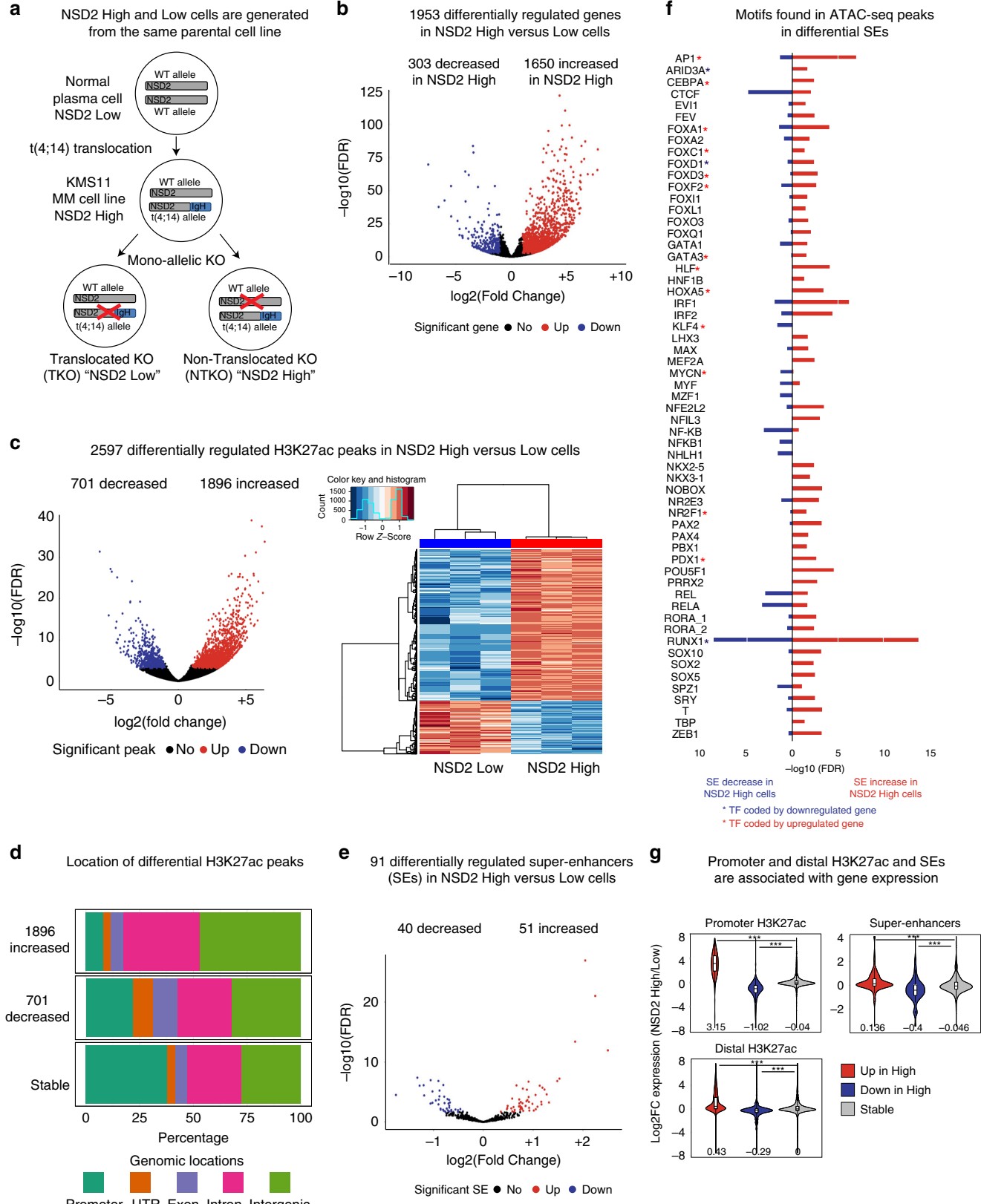

To determine whether spreading of H3K36me2 into intergenic regions could impact the activation status of potential regulatory elements, we analyzed H3K27ac, a feature of both enhancers and promoters. H3K27ac ChIP-seq revealed that a total of 2597 peaks were significantly affected by NSD2 overexpression (1896 increased and 701 decreased peaks) as shown by volcano plots, and heatmaps (Fig. 1c, Supplementary Data 2) and PCA revealed that NSD2 High and Low replicates clustered separately (Supplementary Fig. 2a). Increased and decreased peaks were predominantly located in intergenic and intronic regions

**Fig. 1** NSD2 overexpression leads to alterations in H3K27ac enrichment linked to changes in gene expression. **a** NSD2 Low and High isogenic cell lines generated from the patient-derived KMS11 t(4;14) Multiple Myeloma (MM) cell line: NTKO (non-translocated knockout) and TKO (translocated knockout) cells, which have the translocated allele or the endogenous *NSD2* allele, respectively inactivated. MM multiple myeloma. **b** Volcano plot showing significant NSD2-mediated changes in gene expression ($n = 5$ independent experiments, FDR < 0.01, Wald test). Upregulated genes: 1650 (red, log2-fold change > 1); downregulated genes: 303 (blue, log2-fold change < −1). **c** Violin plot (left panel) and heatmap (right panel) showing significant NSD2-mediated changes in H3K27ac ($n = 3$ independent experiments, FDR < 0.01, Wald test). Decreased peaks: 701 (blue, log2-fold change < −1); increased peaks: 1896 (red, log2-fold change > 1). **d** Genomic locations of the differential H3K27ac peaks. **e** Volcano plot showing significant NSD2-mediated changes in super-enhancers ($n = 3$ independent experiments, FDR < 0.1, Wald test) called based on H3K27ac levels using ROSE. Increased super-enhancers: 51 (red) and decreased super-enhancers: 40 (blue). **f** Transcription factor motifs identified in 119 ATAC-seq peaks of increased (51, red) and decreased (40, blue) super-enhancers using TRAP(−log10 FDR). Red and blue stars indicate that the gene encoding the TF is respectively up-regulated or downregulated in NSD2 High cells. **g** Gene expression changes are associated with H3K27ac changes at promoters, distal H3K27ac and super-enhancers. H3K27ac up, down and stable red, blue, and gray in NSD2 High versus Low cells. Source data are provided as a Source Data file

suggesting that enhancer activity could be affected (Fig. 1d). This pattern was more pronounced for increased peaks.

Previous studies have reported super-enhancer activity changes in MM[32,33]. Super-enhancers are a cluster of enhancers with high levels of activating marks, including H3K27ac[34] that are occupied by lineage-specific master regulator transcription factors (TFs) known to control cell identity[34,35]. Furthermore, super-enhancers are generated at oncogenes and other loci that are important in tumorigenesis[34,35]. To separate super-enhancers from typical enhancers we used 'ROSE' (rank ordering of super-enhancers[34]). We identified 260 and 278 super-enhancers in NSD2 low and high cells, respectively, with 110 overlapping between conditions (Supplementary Fig. 2b). Differential super-enhancers were identified from the union of 428 super-enhancers based on alterations in H3K27ac enrichment. About a third of super-enhancers (91) had significantly altered H3K27ac signal in NSD2 High cells, 51 with increased and 40 with decreased signal (FDR < 0.1; Fig. 1e).

**Changes in H3K27ac are associated with gene deregulation**. To determine which transcription factors (TFs) could bind gained and lost super-enhancers in NSD2 High versus Low cells, we analyzed motifs within 119 and 102 ATAC-seq peaks in the 51 gained and 40 lost super-enhancers, respectively, using transcription factor affinity prediction (TRAP) (Fig. 1e). Some of the motifs were shared between gained and lost super-enhancers (CTCF, RUNX1, etc.), while others were identified in only one category. Furthermore, we found that the expression of some of the genes that encode these TFs increases or decreases (Fig. 1f, red or blue asterisk, respectively). Most of the motifs in super-enhancers were also found in individual gained and lost H3K27ac peaks with the exception of the CTCF motif (Supplementary Fig. 2c). This is consistent with studies showing that super-enhancers are enriched with CTCF more frequently than typical enhancers[36,37].

To examine the connections between alterations in H3K27ac and gene regulation we separated H3K27ac peaks into those located at promoters, non-promoter sites (intragenic and intergenic combined) and super-enhancers. Genomic regions enrichment of annotations tool (GREAT[38]) was used to associate peaks at distal sites and super-enhancers with gene expression changes[39]. H3K27ac peaks at all three sites were significantly correlated with gene-expression changes (Fig. 1g), with the strongest effect seen at promoters. Together these data indicate that overexpression of NSD2 leads to significant changes in H3K27ac linked to deregulation of gene expression.

**Changes in CTCF binding are linked to altered gene expression**. TADs (whose boundaries are enriched for the insulating protein CTCF) have a 3D chromatin structure that has been proposed to delimit the action of enhancers to genes within the same domain[40,41]. Given that marks associated with regulatory elements are significantly altered and correspond to changes in gene expression in NSD2 overexpressing cells, we next asked whether alterations in histone modifications could lead to changes in chromosome organization. We identified a surprising difference in CTCF binding by ChIP-seq: 2197 and 295 CTCF peaks were increased and decreased, respectively, in the NSD2 High cells as shown by volcano plots, and heatmaps (Fig. 2a left and right panels, respectively, Supplementary Data 3). PCA analysis separated NSD2 High and Low replicates into distinct clusters (Supplementary Fig. 3a). Increased CTCF peaks harbor a similar motif to that observed for randomly selected CTCF peaks and the consensus motif MA0139.1 from the Jaspar database (Supplementary Fig. 3b). The increased CTCF peaks were predominantly located in intergenic and intronic regions, as compared with decreased and stable CTCF peaks (Fig. 2b). Moreover, although a large portion of differentially expressed genes are associated with stable CTCF peaks, CTCF peaks that were enriched or depleted at promoters and distal sites were significantly correlated with gene expression changes, with the strongest effect seen at promoters (Fig. 2c). This is consistent with previous studies showing that loss of CTCF at promoters leads to downregulation of genes[8].

**New CTCF and H3K27ac peaks are found in new H3K36me2 domains**. To better understand the links between NSD2 overexpression and the altered genomic profiles of H3K27ac and CTCF, we analyzed the chromatin landscape surrounding the 1000 most enriched new H3K27ac and CTCF peaks in the NSD2 High cells. Importantly, we observed enriched H3K36me2 levels at increased H3K27ac and CTCF peaks as opposed to stable and decreased H2K27ac and CTCF peaks (Fig. 2d and Supplementary Fig. 4a–d). A slight decrease of H3K27me3 was observed at these locations (Fig. 2d) suggesting that increases in H3K27ac peaks may be partially dependent on H3K27me3 changes. New H3K27ac and CTCF peaks were also associated with a gain in overlapping ATAC-seq peaks while decreased peaks show a slight reduction in chromatin accessibility and stable peaks show stable accessibility (Fig. 2d and Supplementary Fig. 4a–d). Although randomly selected sites show an enrichment of H3K36me2, this is not sufficient to drive chromatin accessibility and CTCF or H3K27ac enrichment (Supplementary Fig. 4e, f). This data suggests that spreading of H3K36me2 provides a more accessible chromatin landscape favorable for CTCF and H3K27ac enrichment at a subset of loci. As shown in Fig. 2e, these changes are linked to the activation of oncogenes involved in MM including *SYK*[42–44], *MET*[45,46], and *SH3GL3*[47] in NSD2-overexpressing cells (Fig. 2e). Of note, few changes were observed for the 1000 most stable and decreased H3K27ac and CTCF peaks, but a decrease in both H3K27ac and CTCF was linked to a slight increase in H3K27me3 (Supplementary Fig. 4).

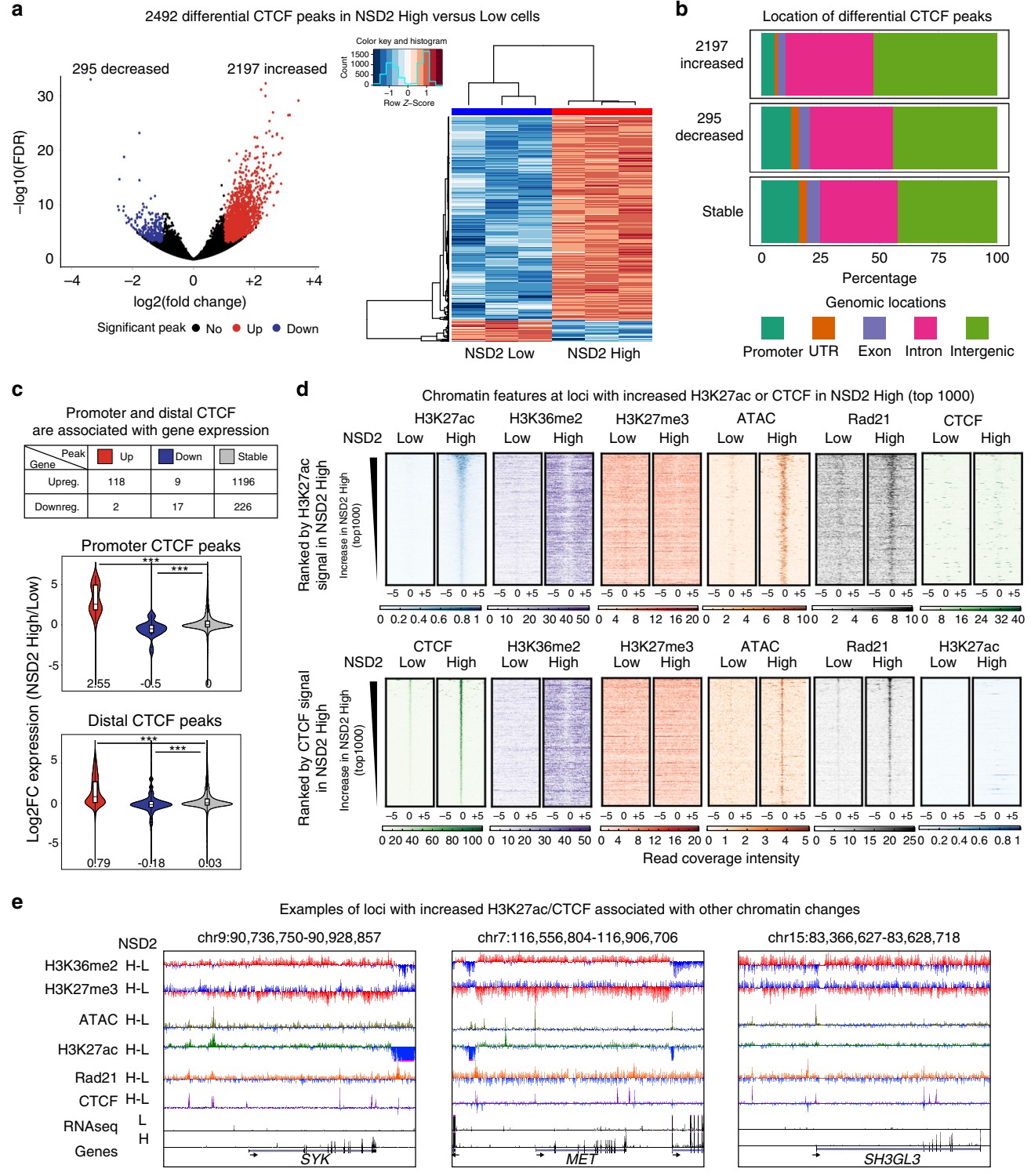

**Fig. 2** New CTCF and H3K27ac peaks are located within expanded H3K36me2 domains. **a** Violin plot (left panel) and heatmap (right panel) showing significant NSD2-mediated changes in CTCF binding ($n = 3$ independent experiments, FDR < 0.01, Wald test). Increasing peaks: 1650 (red, log2-fold change > 1); decreasing peaks: (blue, log2-fold change < −1). **b** Genomic locations of differential CTCF peaks using ChIPseeker (promoter: −/+3kb around TSS). **c** Gene expression changes are associated with CTCF changes (upreg. and downreg.: upregulated and downregulated genes), at promoters (middle panel) and distal CTCF sites (bottom panel). CTCF up, down and stable are, respectively, shown as red, blue and gray in NSD2 High versus Low cells. Source data are provided as a Source Data file. **d** Heatmaps of H3K27ac, H3K36me2, H3K27me3, ATAC-seq, Rad21, and CTCF signal at the top 1000 increased H3K27ac and CTCF peaks in NSD2 High cells (top and bottom panels, respectively). Peaks were ranked by H3K27ac and CTCF signal in NSD2 High cells. **e** UCSC genome browser screenshots of chromatin features at regions surrounding three oncogenes (*SYK*, left panel; *MET*, middle panel; and *SH3GL3*, right panel). H-L refers to subtraction of the ChIP-seq signal (NSD2 High−Low)

Cohesin's importance in chromatin looping has been widely demonstrated[9–11,48], with loops separating into CTCF-dependent and CTCF-independent categories. Specifically, CTCF-independent cohesin-mediated loops are cell type specific, dynamic loops that form between enhancers and promoters and involve cell type specific TFs[49]. To investigate whether new H3K27ac and CTCF peaks had the potential to be involved in looping in the NSD2 High cells we performed Rad21 ChIP-seq. Interestingly, new Rad21 peaks were associated with both enriched H3K27ac and CTCF sites in the NSD2-overexpressing cells. However, gained CTCF and H3K27ac peaks did not overlap with each other (Fig. 2d), suggesting that loops involving new regulatory elements associated with enriched H3K27ac were generated in a CTCF independent manner.

**NSD2 overexpression drives A/B compartment switching**. Given that significant changes in CTCF or H3K27ac peaks overlap Rad21 alterations in NSD2 High cells, we next asked whether 3D organization was altered. We performed Hi-C in duplicate using the Arima Kit and processed the data using Hi-C bench[50]. The number of valid read pairs was consistent between replicates (~120 million to ~180 million (Supplementary Fig. 5a)) and the PCA showed that NSD2 High versus Low replicates separated as expected (Supplementary Fig. 5b). Active A and Inactive B compartment status was analyzed in merged NSD2 Low and High replicates using the Eigen vector (principal component 1, PC1, see the "Methods" section for details)[1]. We observed significant compartment weakening in NSD2 High cells (Supplementary Fig. 5c), as exemplified by the changes visualized on chromosome 7 in Fig. 3a. Overall, the number of A and B compartments was lower in NSD2 High versus Low cells and we detected 324 regions that switched from A to B and 491 regions that switched from B to A (Fig. 3b–d and Supplementary Fig. 5d). Although there were more regions switching from B to A, these regions were smaller and the portion of the genome was comparable to the regions switching from A to B (Supplementary Fig. 5e–g). Compartment switching from B to A overlapped with increased H3K36me2 levels and a reduction in H3K27me3, while the reverse was true for switching from compartment A to B (Fig. 3d, e and Supplementary Fig. 5h). Of note, an increase in H3K36me2 is not limited to switching regions (Supplementary Fig. 5h). We speculate that an overall increase in CTCF peaks in the genome in NSD2 High cells could contribute to weakening of compartment structure, consistent with previous findings showing that loss of cohesin and TAD structure strengthens compartmentalization[9,10].

**Changes in TAD boundaries and intra-TAD interactions**. Both Hi-C replicates from each condition harbored similar numbers and sizes of TADs (Supplementary Fig. 6a, b) and were merged for TAD analysis to determine the impact of NSD2 overexpression on 3D genome organization. Using Hi-C bench, we assigned a mean boundary score to each replicate using a resolution of 40 kb[50] to determine if there were changes in TAD boundaries (see the "Methods" section for details and Supplementary Fig. 6c). NSD2 overexpression was associated with a significant (FDR < 0.05) increase in insulation at TAD boundaries (red, 61) and much fewer boundaries that were weakened (blue, 5) (Fig. 4a), synonymous with changes in CTCF and Rad21 binding (Fig. 4b). H3K36me2 levels are higher in NSD2 High versus Low cells both at boundaries and surrounding regions. However, we could not detect any difference in H3K36me2 enrichment in the NSD2 high cells at boundaries with increased strength versus stable boundaries (Supplementary Fig. 6d). In addition, we found that NSD2 High cells were predominantly

associated with significant gains (red, 229) and fewer decreases in intra-TAD interactions (blue, 30) (Fig. 4c). Increased intra-TAD interactions were linked to expanded H3K36me2 domains and a reduction in H3K27me3, while decreases in intra-TAD interactions were linked to a reduction in H3K36me2 (Fig. 4d).

As mentioned above, NSD2 overexpression was associated with a significant strengthening of TAD boundaries concordant with increases in CTCF and Rad21 binding (Fig. 4b) as well as transcriptional activation. As an example we highlight activation of *FZD8* (log2 fold-change = 3.16, FDR < 0.01), a gene that is involved in the WNT-signaling pathway[51] (Fig. 4e). *FZD8* is located at a CTCF-enriched boundary making a new contact with a downstream region that gains a CTCF/Rad21 site (blue stripe in Fig. 4e) in NSD2 High cells. This is demonstrated by an increase in boundary score (arrows in Fig. 4e, f) and the formation of a new Hi-C loop (circle in Fig. 4f). Consistent with the data in Fig. 2d, the whole domain overlaps a region that is enriched for H3K36me2 and within the new loop there is increased H3K27ac.

Previous studies have proposed that *FGF13* expression enables cells to cope more effectively with RAS-mediated stress and oncogene-mediated excessive protein synthesis, which is known to occur in MM in the form of antibody production[52]. In Supplementary Fig. 7a, b we show that activation of *FGF13* occurs at a region with an altered TAD boundary and increased intra-TAD interactions. This region harbors newly formed H3K27ac peaks, enriched chromatin accessibility, and overlaps a domain enriched for H3K36me2 that undergoes B to A compartment switching. Increased CTCF and Rad21 binding are at the edges of increased Hi-C loops, suggesting that newly formed insulated neighborhoods could facilitate the interaction between *FGF13* and regulatory elements.

We also found contacts between H3K27ac-enriched regions within domains bound by enriched CTCF. As an example, we highlight the *KRAS* oncogene (yellow strip Supplementary Fig. 7c, d) that plays an important role in driving the MM phenotype[53]. As shown in Supplementary Fig. 7b, upregulation of *KRAS* (log2-fold change = 0.86, FDR < 0.01) coincides with a loop between *KRAS* and a super enhancer, whose activity is enriched in NSD2 high cells (see arrow and blue stripe Supplementary Fig. 7c and arrows and circle in Supplementary Fig. 7d), and an increase in intra-TAD activity. The above examples indicate that significant changes in H3K27ac, CTCF and transcriptional output are frequently clustered together.

**Clustered chromatin and transcription change**. To investigate the connections between alterations in gene expression, CTCF, and H3K27ac from a global perspective, we focused our analysis on TADs and CTCF-mediated loops identified by CTCF HiChIP in NSD2 High and Low cells[54]. Statistical analysis of HiChIP data was carried out using FitHiChIP[55] (see the "Methods" section). CTCF-mediated interactions were considered significant using an FDR < 0.01.

To determine whether significant changes in CTCF, H3K27ac, and gene expression in NSD2 High were localized together we focused on common TADs and CTCF-mediated loops (see details in the "Methods" section). This allowed us to reliably compare between the NSD2 High versus Low condition. The number of significant changes in CTCF, H3K27ac, and gene expression in these common domains is shown in Supplementary Fig. 8a, b. TADs and CTCF loops had a median size of 600 and 210 kb, respectively (Supplementary Fig. 8c).

As shown in the scheme in Fig. 5a, we selectively analyzed TADs and CTCF-mediated loops containing at least one CTCF and H3K27ac peak and one transcriptionally active gene in either the NSD2 Low or High condition. The three variables are

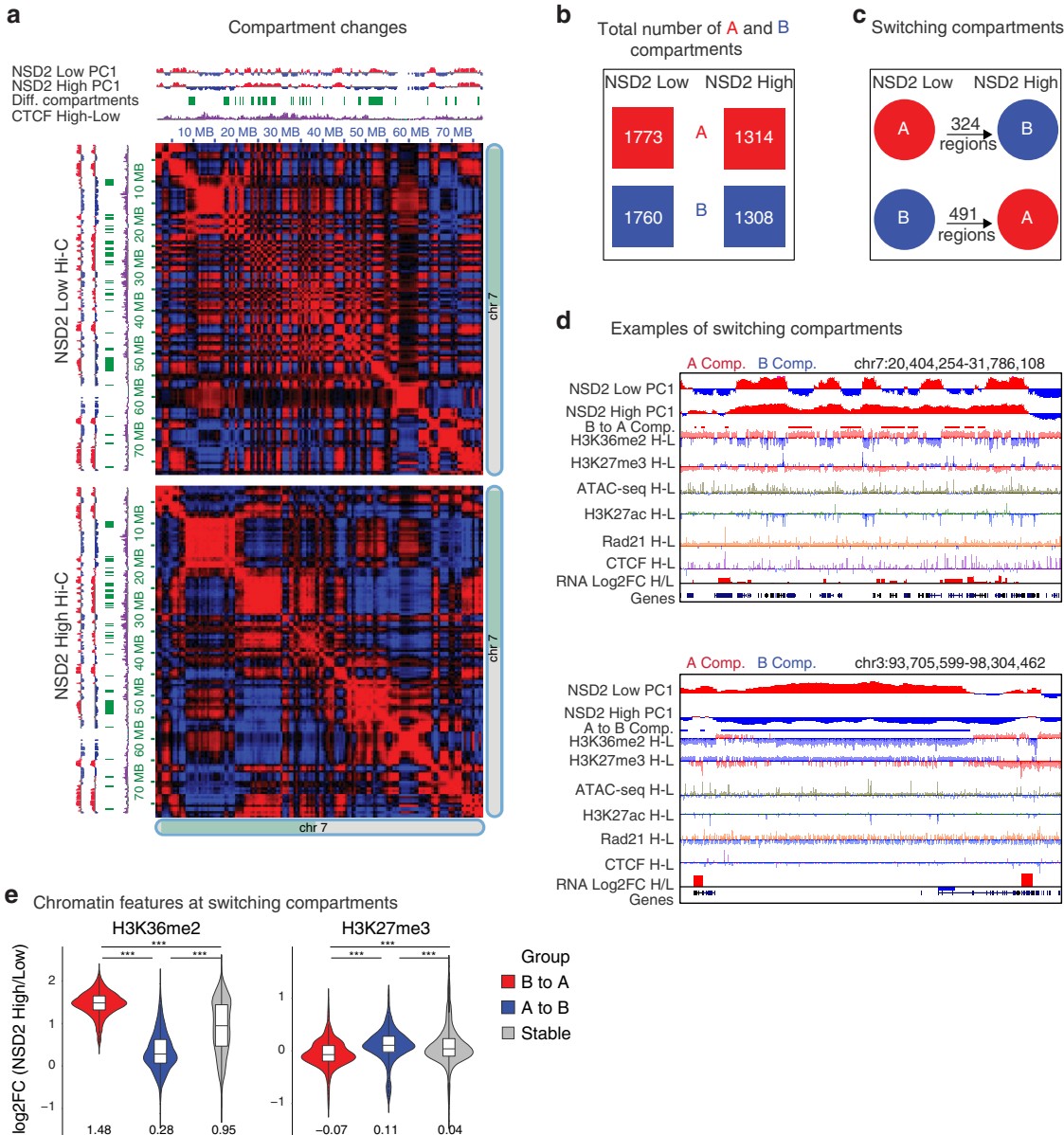

**Fig. 3** NSD2 overexpression drives A/B compartment switching. **a** Compartment weakening for half of chromosome 7 is shown in NSD2 High versus Low cells. Top and left: Eigen vector (PC1) for compartments A and B in red and blue, respectively. Switching regions are shown in green and in purple the subtraction of CTCF signal (NSD2 High–Low). Heatmaps represent the Pearson correlation of interactions in NSD2 Low (top) and High (bottom) cells. Positive and negative Pearson correlations between two loci are represented in red and blue, respectively. **b** Total number of A (red) and B (blue) compartments in NSD2 High versus Low cells. **c** Number of regions that switch compartments from B to A in (491 regions) or A to B (324 regions) in NSD2 High versus Low cells. **d** IGV screenshots show examples of regions that switch from A to B (top panel) and B to A (bottom panel) in NSD2 High versus Low cells. Eigen vectors (PC1), differential compartment switching, subtraction tracks of H3K36me2 and H3K27ac and expression for NSD2 High and Low cells are shown. Regions that significantly switch from B to A and from A to B are indicated in red (top panel, B to A) or in blue (bottom panel, A to B), respectively. H–L refers to subtraction of the ChIP-seq signal (NSD2 High–Low). **e** Changes in H3K36me2 and H3K27me3 levels within regions that switch compartment from B to A (red), A to B (blue), or are stable (gray) in NSD2 High cells. Log2-fold changes (NSD2 High versus Low cells) for H3K36me2 and H3K27me3 are shown. The median is indicated under the violins. Source data are provided as a Source Data file

positively correlated in both TADs and CTCF-mediated loops, without considering any fold change or FDR thresholds (Supplementary Fig. 9a, b). Positively correlated features are shown by pairwise and three-way log2 fold change comparisons in the 2D and 3D plots, respectively. We also included a pairwise and three-way comparison of intra-TAD interaction changes and PC1 changes to analyze compartment changes within insulated regions that contained alterations in CTCF (see the "Methods" section for details), H3K27ac, and transcriptional output (Supplementary Fig. 9a). Again, we found that alterations in

intra-TAD interactions and compartment switching were concordant within TADs with alterations in CTCF, H3K27ac, and transcriptional output. The correlation improved when we restricted our analyses to significant changes in CTCF, H3K27ac, and transcriptional output within TADs and CTCF-mediated loops (Fig. 5b, c). We also confirmed that high PC1 mean difference values were associated with regions switching from B to A using the Homer algorithm. We found 83% of TADs with B to A switching regions (orange dots) overlapped with active features in the 3D graph in common TADs with

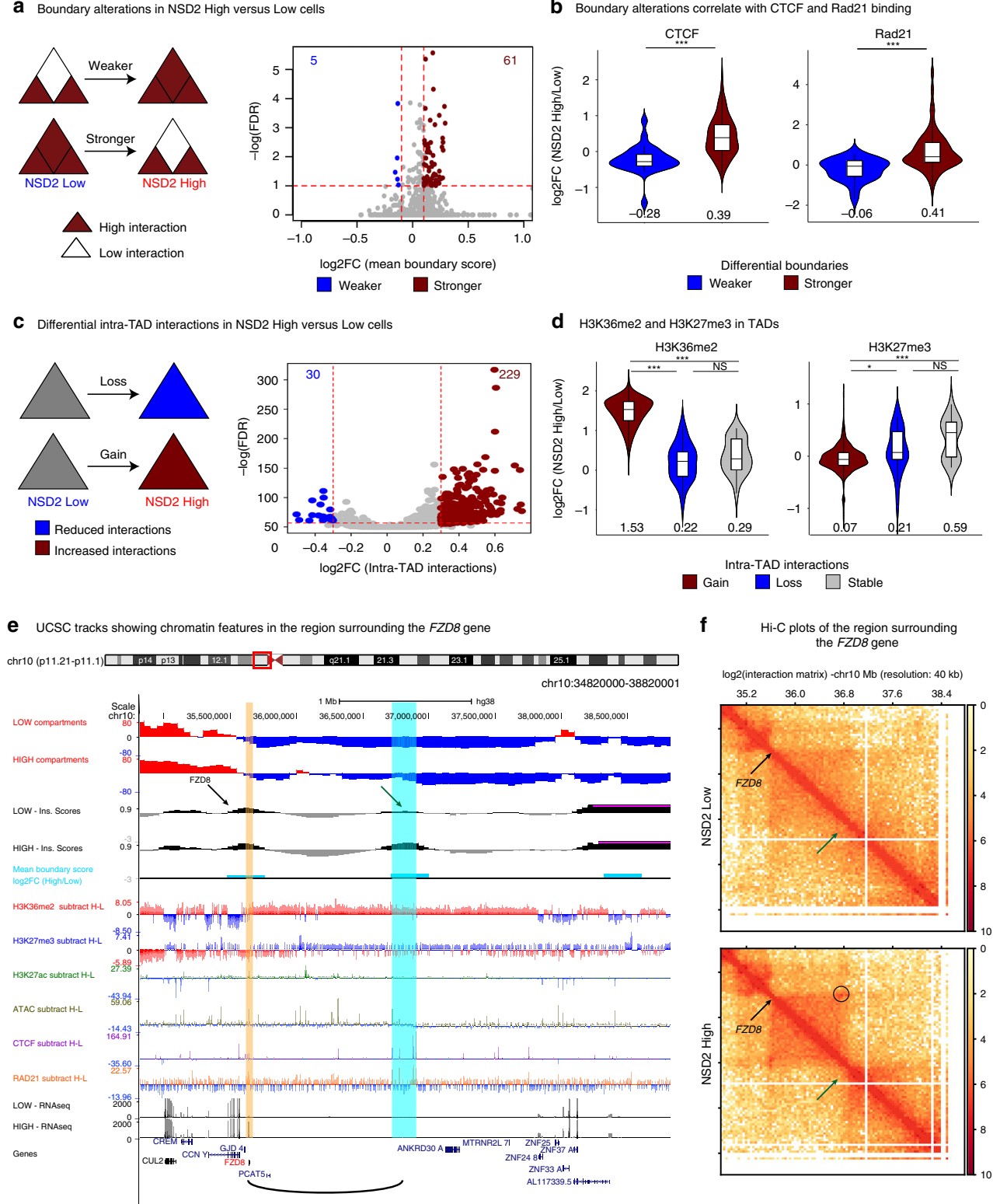

**a** Boundary alterations in NSD2 High versus Low cells

**b** Boundary alterations correlate with CTCF and Rad21 binding

**c** Differential intra-TAD interactions in NSD2 High versus Low cells

**d** H3K36me2 and H3K27me3 in TADs

**e** UCSC tracks showing chromatin features in the region surrounding the *FZD8* gene

**f** Hi-C plots of the region surrounding the *FZD8* gene

differential changes in CTCF, H3K27ac, and RNA (25% of the total positively correlated active TADs) (Fig. 5b). The relationship between significant changes in gene expression and concordant versus discordant alterations in H3K27ac and CTCF in TADs and CTCF-mediated loops is shown in Supplementary Fig. 9c. These analyses indicate that the vast majority of changes in CTCF, H3K27ac, and transcriptional output are positively correlated.

**Altered chromatin drives oncogenic transcriptional programs.** Within the TADs and loops that harbor significant changes in CTCF, H3K27ac, and transcriptional output we detected several newly activated oncogenes associated with MM-related pathways. These include the protein tyrosine phosphatase gene, *PTPN13*[56], *FGF13*[57], and *ETV5*[58]. The *PTPN13* gene, which regulates cell growth[59], differentiation[56], mitotic cycle, and oncogenic transformation is located within a CTCF-mediated loop which harbors

**Fig. 4** Changes in TAD boundaries and intra-TAD interactions. **a** Boundary alterations in NSD2 High versus Low cells. NSD2 overexpression is associated with increases (red, 61) and decreases (blue, 5) in TAD boundary strength ($n = 2$ independent experiments, cutoffs of absolute Log2-fold change > 0.1 and FDR < 0.05). **b** TAD boundary increases and decreases in NSD2 High versus Low cells are associated with increases and decreases in CTCF and Rad21 binding, respectively. The median is indicated under the violins. **c** Intra-TAD interaction changes in NSD2 High versus Low cells for overlapping TADs (1564). NSD2 overexpression is associated with gain (red, 229) and loss (blue, 30) of intra-TAD interactions ($n = 2$ independent experiments, cutoffs of absolute Log2-fold change > 0.3 and FDR < 0.05). **d** Changes in H3K36me2 and H3K27me3 within TADs that have increased (red), decreased (red), or stable (gray) interactions in NSD2 High cells. Log2-Fold changes (NSD2 High/Low cells) for H3K36me2 and H3K27me3 ChIP-seq are shown. The median is indicated under the violins. **e** UCSC tracks showing chromatin features in the region surrounding the *FZD8* gene (*FZD8* gene indicated in red and location highlighted by a yellow stripe) and the new contact that is formed by a strengthened boundary (blue stripe). The graphical representation of interaction between *FZD8* and the super enhancer is shown as a loop below. H–L refers to subtraction of the ChIP-seq signal (NSD2 High–Low). **f** Hi-C plots of the region surrounding the *FZD8* gene. Top panel: NSD2 Low, bottom panel: NSD2 High. Green arrow identifies the TAD boundary that is strengthened in NSD2 High versus Low cells. Black arrow indicates the *FZD8* gene. Circle indicates a new loop between *FZD8* and the boundary. Source data for violin plots in panels **b** and **d** are provided as a Source Data file

significant changes in CTCF and H3K27ac (Fig. 6a left), and within a region that undergoes B to A compartment switching. *ETV5*, a gene involved in the KRAS pathway[60], is another example of gene activation that occurs in a CTCF-mediated loop and a TAD that harbors significant changes in CTCF and H3K27ac (Fig. 6a right). As with *PTPN13*, the insulated region contains new H3K27ac and CTCF peaks, overlaps a region that has switched from compartment B to A and in addition has increased intra-TAD interactions (Fig. 6a right).

*SYK* is another example of an oncogene that is activated within a CTCF-mediated loop that has significant increases in H3K27ac and CTCF peaks. To investigate whether upregulation of *SYK* in NSD2 High cells involves alterations in promoter contacts, we performed high resolution 4C-seq from a viewpoint located on a CTCF site 27 kb upstream of the *SYK* promoter. We identified significantly increased interactions with upstream and downstream regions (FDR < 0.01; marked by the small dots on the 4C plot in Fig. 6b). All of these sites correspond to enriched CTCF or H3K27ac peaks that overlap with enriched cohesin peaks in the NSD2 High cells (Fig. 6c). The interactions identified by 4C-seq can also be visualized by Hi-C (arrows and circles in Fig. 6d). Interestingly, the change in interactions seen in Fig. 6b overlap with a region that changes from compartment B to A and has increased intra-TAD interactions. As in other examples, these changes occur in a domain where H3K36me2 is enriched in the NSD2 High cells. Taken together our data indicate that gene expression changes and activation of oncogenes in NSD2 overexpressing cells are found in insulated regions with accompanying alterations in H3K27ac and CTCF peaks.

**Links between chromatin and transcriptional changes**. To better understand the relationship between transcriptional changes and chromatin changes in TADs and CTCF loops we modeled the probability that a gene is differentially expressed, as a function of changes in CTCF and/or H3K27ac within these insulated domains, using logistic regression. First, we assigned all genes, CTCF, and H3K27ac peaks to all TADs and CTCF loops in the NSD2 High and Low conditions. Then, we aggregated these and computed whether a gene shares a TAD or CTCF loop with a differential CTCF and/or H3K27ac peak. Finally, we checked whether changes in CTCF or H3K27ac peaks between the two conditions in the same TAD or CTCF loop have an effect on whether a gene is differentially expressed (Fig. 7a). We found that differentially expressed genes are significantly enriched within the same CTCF loop as altered H3K27ac peaks. Furthermore, differentially expressed genes are significantly enriched within the same TAD as altered H3K27ac and CTCF peaks. We tested for possible synergistic or antagonistic effects of H3K27ac and CTCF peaks and neither was significant. These changes represent 60.4% of the differentially expressed genes. Changes in gene expression

cluster with changes in CTCF and/or H3K27ac in only a subset of TADs (17%) and loops (13.8%). The number of CTCF and H3K27ac changes that occur within TADs and loops that have altered gene expression is shown in Fig. 7b.

**Discussion**

Deletion of individual CTCF sites is known to disrupt chromosome interactions allowing the spreading of active chromatin and altering gene regulation[18–20]. However, it is not known whether alterations in chromatin can affect CTCF binding, 3D genome organization and gene expression. To address this question we examined the effect of NSD2 overexpression in t(4;14) MM. The activation of oncogenic transcriptional pathways in this context has been shown to rely on the histone methyl transferase activity of NSD2 and deposition of H3K36me2[31]. Here we show that NSD2-overexpressing cells coopt altered chromatin domains to drive changes in 3D organization including A/B compartmentalization and TAD architecture, linked to the activation of oncogenic transcriptional pathways.

Expansion of H3K36me2 outside of active gene bodies provides a favorable environment for H3K27ac enrichment and CTCF binding. Hi-C and HiChIP analyses reveal that changes in H3K27ac, CTCF, and transcriptional activity occur in a predominantly concordant manner in common TADs (72%) and CTCF loops (82%). Furthermore, intra-TAD interactions and B to A compartment changes are positively correlated with changes in H3K27ac, CTCF, and transcriptional activity in TADs. Importantly, we find that 60.4% of the deregulated genes cluster together in a subset of TADs (17%) or CTCF-mediated loops (13.8%) that harbor significant changes in CTCF and/or H3K27ac binding. Using a logistic regression model, we demonstrate that differentially expressed genes are significantly enriched within the same insulated domain as altered H3K27ac and CTCF peaks. Our findings are consistent with a model in which gene regulation is constrained at the 3D level by insulated boundaries that restrict the influence of enhancers. As highlighted in the examples in this manuscript, these changes provide an explanation for the activation of many oncogenes associated with MM.

What is the relationship between alterations in H3K36me2, H3K27ac, CTCF, and A/B compartment changes? Since compartment switching is linked to changes in chromatin activity it is possible that changes in H3K36me2 on its own could induce compartment changes. It would be difficult to tease this apart because spreading of H3K36me2 into intergenic regions is accompanied by changes in transcriptional activity, H3K27ac, CTCF, ATAC-seq, and Rad21, all of which could contribute to changes in compartmentalization. We speculate however that it is the overall increases in CTCF/Rad 21 binding that plays a key role in compartment weakening, in line with the finding that acute depletion of cohesin and loss of TAD structure leads to

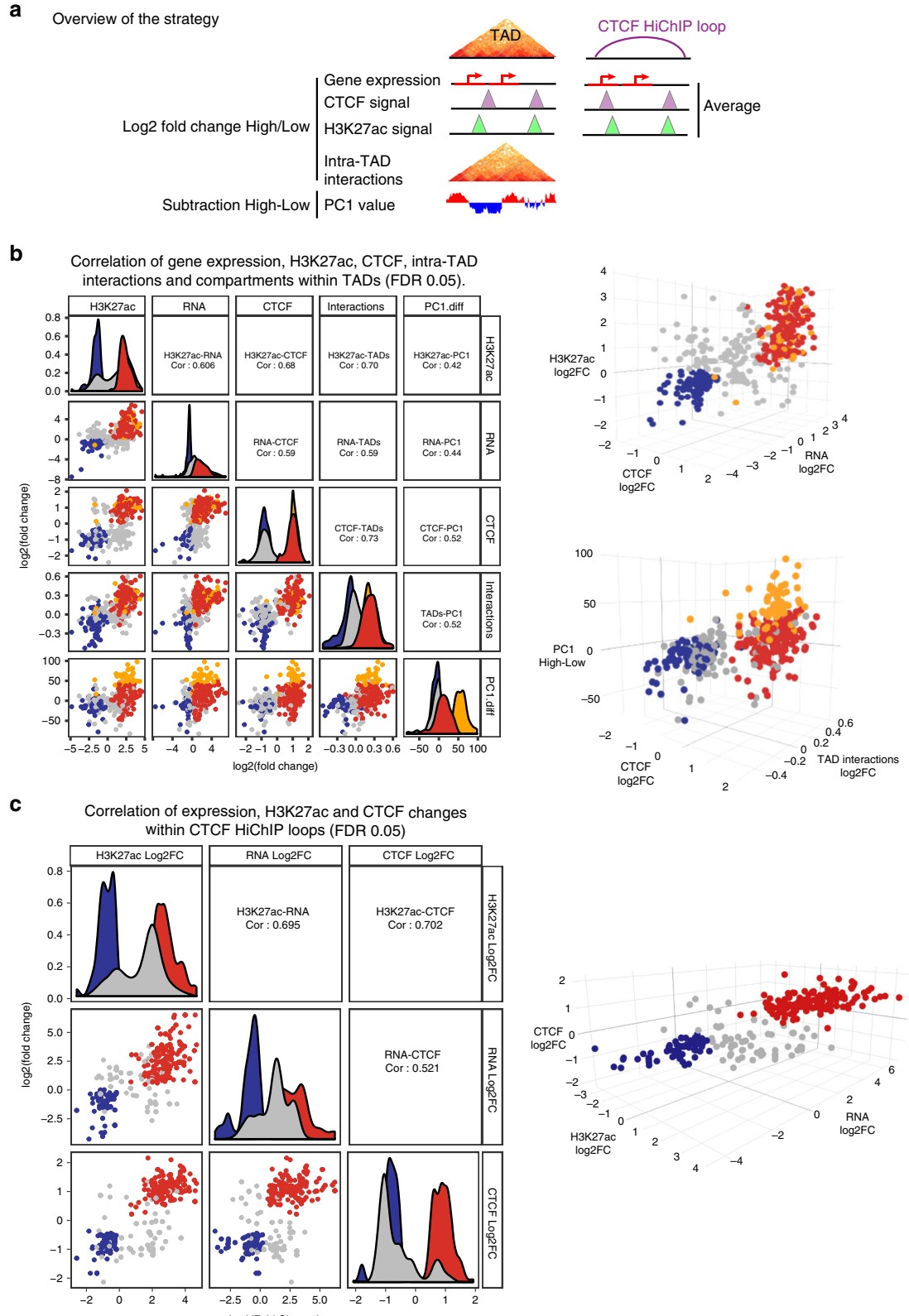

strengthening of compartments[9,10]. Thus, although compartments and TADs are formed through independent pathways, TAD structures interfere with compartmentalization.

Since the NSD2 phenotype relies on the histone methyl transferase activity of the protein, as previously described[23,25], the upstream causal event is the expansion of H3K36me2 domains. Beyond this change it is not clear what order the other events

occur in, or whether they rely on each other. The causal directionality would be difficult to determine as we could not exclude the possibility that chromatin changes are inter-dependent. It is unlikely that alterations in H3K27ac enrichment and CTCF binding are dependent on each other as they do not overlap. We speculate that opening up of the chromatin through deposition of the active mark, H3K36me2 allows binding of transcription

**Fig. 5** NSD2 overexpression drives concordant chromatin and transcriptional changes in insulated domains. **a** Scheme illustrating the strategy to identify chromatin and transcriptional changes within TADs or CTCF HiChIP loops that were filtered to have at least one differentially expressed gene, CTCF and H3K27ac peak. **b** Pairwise (2D scatter plots left panel) and three-way (3D scatter plots right panels) comparisons representing significant log2-fold-changes in gene expression, H3K27ac, CTCF, intra-TAD interactions, and PC1 values (representing subtraction of NSD2 High and Low levels) within TADs that have at least one significantly differentially expressed gene, CTCF and H3K27ac peak (FDR < 0.05). Concordant increased and decreased changing TADs are colored in red and blue, respectively. TADs that switch from B to A according to HOMER analysis (see the "Methods" section for details) are highlighted in orange. Pearson correlations are indicated. **c** Pairwise (2D scatter plots left panel) and three-way (3D scatter plots right panels) comparisons representing significant log2-fold changes of NSD2 High versus Low levels in gene expression, H3K27ac and CTCF within CTCF HiChIP loops that have at least one differentially expressed gene, CTCF and H3K27ac peak (FDR < 0.05). Concordant increased and decreased changing loops are colored in red and blue, respectively. Pearson correlations are indicated. Source data are provided as a Source Data file for **b** and **c** panels

factors, such as AP-1, which recruits CBP/p300[61] that mediates deposition of H3K27ac, a well-defined marker of enhancer activity (Fig. 7c). Indeed, our studies reveal that the motif for transcription factors, such as FOXA1, CEBPA, and AP-1 are enriched at increased H3K27ac peaks, including those designated as super-enhancers (Supplementary Fig. 2c, d). Moreover, expression of *FOS2L* and *JUN* that encode the AP-1 subunits are upregulated in NSD2 high cells. AP-1 is enriched in activation hubs that connect enhancers and promoters in macrophages[62], and it may participate in regulating enhancer–promoter contacts in NSD2 overexpressing MM cells. Similarly, we speculate that H3K36me2-mediated accessibility promotes binding of CTCF. H3K27ac enriched regulatory elements and CTCF bound sites are known to participate in cohesin-mediated loop formation and in line with this, we show that new H3K27ac and CTCF peaks overlap with new Rad21 peaks and form new or strengthened contacts linked to changes in gene expression. An overall increase in CTCF/Rad21 and H3K27ac, as well as transcriptional activity likely contributes to the overall increase in intra-TAD interactions as shown by the pairwise and three way-comparisons in Fig. 5. These changes are also positively correlated with B to A compartment changes. Specifically, we find that 85% of regions that switch from B to A are found in TADs with a significant increase in CTCF, H3K27ac, and transcriptional output and perhaps not surprisingly, compartment switching occurs in TADs with the highest activity of all three.

Here we have identified a functional interplay between NSD2-mediated changes in chromatin, 3D organization and transcriptional output. These results highlight the bidirectional relationship between chromatin status and 3D genome organization in gene regulation. In the context of NSD2 overexpression in MM, we demonstrate that alterations in CTCF and H3K27ac increase the probability of changing gene expression within the same TAD or CTCF loop. These chromatin alterations are associated with the majority of changes in gene expression including genes, such as *KRAS* and *SYK*, which in the absence of driver mutations contribute to the oncogenic program of the disease as highlighted by the enrichment of KRAS pathway genes amongst deregulated genes (Supplementary Fig. 1d). This work demonstrates that alterations in chromatin modifiers in disease states enables dissection of the interplay between 2D and 3D chromatin structure and their links to gene regulation.

## Methods

**Cell lines.** NSD2 High (one clone of NTKO and KMS11 parental cell line, which is a human myeloma cell line from a female patient) and NSD2 Low cells (two clones of TKO) were obtained from Ben Ho Park[24] who generated the cell lines. Upon reception, the phenotype of the cells was as described[24], and NSD2 High and Low cells harbored expected alterations in NSD2 RNA and protein as confirmed by RNA-seq and Western-blot, respectively. Levels of H3K36me2 and H3K27me3 have also been assessed by Western-blot (Supplementary Fig. 1c and Source Data file). Additional cell authentication was not performed. Cells were maintained in culture as per Ben Ho Park's recommendations. Briefly, cells were grown in RPMI-1640 supplemented with 10% fetal bovine serum, 100 Units per ml penicillin and 100 μg/ml streptomycin and subcultured twice a week by dilution. Our PCA

revealed that NTKO (two replicates from independent cultures for CTCF and H3K27ac ChIP-seq, and four replicates from independent cultures for RNAseq) and KMS11 (one replicate) were similar in terms of gene expression, CTCF and H3K27ac peaks, as well as TKO clones and replicates. Information about cell lines, replicates and respective use in experiments is available in Supplementary Table 1. For downstream experiments, NTKO and KMS11 cells were trypsinized while TKO cells were directly obtained from suspension cultures. Cells were freshly crosslinked for ChIP-seq, Hi-C, Hi-ChIP, and 4C (see corresponding sections for detailed protocols), resuspended in RLT buffer for RNA extraction as per the instructions provided by the kit (RNeasy plus kit from QIAGEN), or directly processed for ATAC-seq (see corresponding section for detailed protocol). For western blotting, the following antibodies have been used: anti-H3 (Abcam, #ab1791, diluted 1:20,000), anti-WHSC1/NSD2 (Abcam Cat #ab75359, diluted 1:2000), anti-H3K27me3 (Millipore Cat#07-449, dilution 1:20,000), anti-H3K36me2 (Active Motif Cat# 39255, dilution 1:2000).

**RNA-seq experimental procedures.** RNA was extracted from five replicates for each condition (for NSD2 High: one KMS11 and four NTKO replicates from independent cultures, for NSD2 Low, two replicates from one clone, and three replicates from the other one, from independent cultures), using the RNeasy plus kit from QIAGEN. Poly-adenylated transcripts were positively selected using the NEBNext® Polya mRNA Magnetic Isolation Module following the kit procedure. Libraries were prepared according to the directional RNA-seq dUTP method adapted from http://waspsystem.einstein.yu.edu/wiki/index.php/Protocol:directional_WholeTranscript_seq that preserves information about transcriptional direction. Sequencing was performed with Illumina Hi-Seq 2500 using 50 cycles paired-end mode.

**ATAC-seq experimental procedures.** NTKO cells were trypsinized while TKO cells were directly obtained from suspension cultures. Cell were counted to collect 50,000 cells per replicate. Two different cultures of NTKO cells were used for the NSD2 High condition and one culture from two different clones of TKO for the NSD2 Low condition. The procedure was repeated on three independent days for a total of six replicates for NSD2 High and six replicates for NSD2 Low cells. The assay was performed as described previously[63]. Cells were washed in cold PBS and resuspended in 50 μl of cold lysis buffer (10 mM Tris–HCl, pH 7.4, 10 mM NaCl, 3mM MgCl2, 0.1% IGEPAL CA-630). The tagmentation reaction was performed in 25 μl of TD buffer (Illumina Cat #FC-121-1030), 2.5 μl Nextera Tn5 Transposase, and 22.5 μl of Nuclease Free H2O at 37 ℃ for 30 min. DNA was purified on a column with the Qiagen Mini Elute kit, eluted in 10 μl H2O. Purified DNA (10 μl) was combined with 10 μl of H2O, 2.5 μl of each primer at 25 mM and 25 μl of NEB Next PCR master mix. DNA was amplified for five cycles and a monitored quantitative PCR was performed to determine the number of extra cycles needed as per the original ATAC-seq protocol[63]. DNA was purified on a column with the Qiagen Mini Elute kit. Samples were quantified with the Tapestation bioanalyzer (Agilent) and the KAPA Library Quantification Kit and sequenced on the Illumina Hi-Seq 2000 using 50 cycles paired-end mode. The six replicates were sequenced independently and three replicates were pooled together during the data processing in order to get two replicates with sufficient sequencing depth for downstream analysis (see ATAC-seq processing data for details).

**ChIPmentation experimental procedures.** H3K27ac and CTCF ChIP-seq were performed in triplicate (for NSD2 High: one KMS11 and two NTKO replicates from independent cultures, for NSD2 Low, two replicates from one clone and one replicate from the other one, from independent cultures). H3K36me2 and Rad21 ChIP-seq were performed in duplicate (for NSD2 High: one KMS11 and one NTKO replicate for H3K36me2 and two NTKO replicates from independent cultures for Rad21, for NSD2 Low: one replicate from each clone). Cells were fixed in culture medium in 1% formaldehyde at RT for 10 min and quenched with 0.125 M glycine. Pellets were washed twice with ice-cold PBS, snap-frozen in liquid nitrogen and stored at −80 ℃. ChIP-seq was performed as per the original ChIPmentation protocol[64] in triplicate for CTCF and H3K27ac, and in duplicate for H3K36me2 and Rad21. Briefly, chromatin was lysed during a 10 min rotation in the cold room in 350 μl of lysis buffer (10 mM Tris–HCl pH 8.0, 100 mM NaCl,

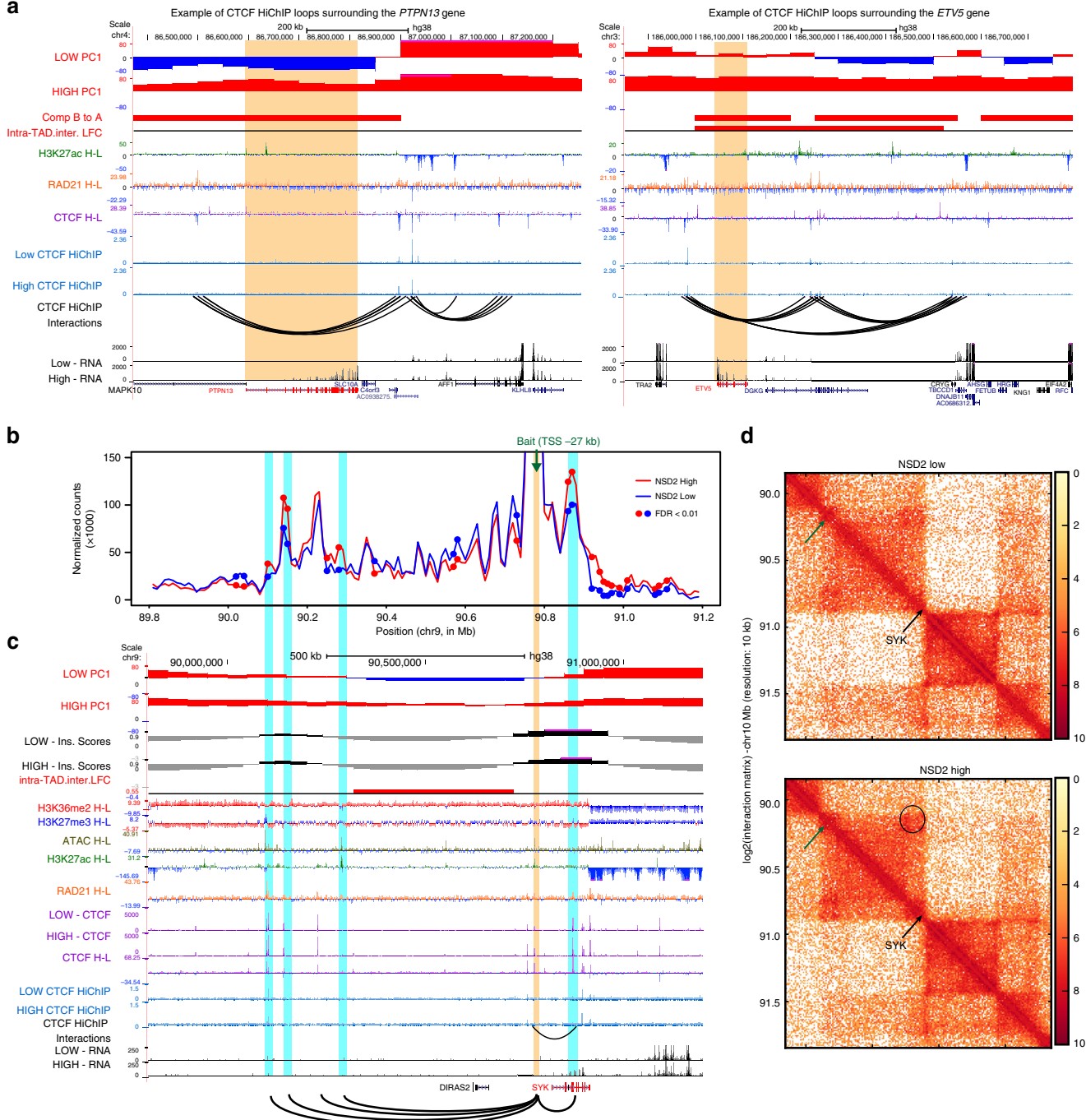

**Fig. 6** Cells co-opt altered chromatin domains to drive oncogenic transcriptional programs. **a** UCSC tracks showing chromatin features and CTCF-mediated loop changes in the region surrounding the *PTPN13* (left panel) and ETV5 (right panel) genes (*PTPN13* and ETV5 genes are indicated in red and highlighted with a yellow stripe). H−L refers to subtraction of the ChIP-seq signal (NSD2 High−Low). **b** Interaction profile of a 4C bait located 27 kb downstream of the *SYK* promoter (green arrow, bait −27 kb) in a 2.4 Mb region surrounding the *SYK* gene in NSD2 High (red line represents the average between two replicates) and Low (blue line represents the average between two replicates) cells using 4C counts in 20 kb sliding windows. DESeq2 analysis identified significantly different 4C signal in duplicated 4C samples from NSD2 High versus Low cells. Regions with differential interactions are indicated by red and blue dots (*n* = 2 independent experiments, FDR < 0.01). **c** UCSC tracks showing chromatin features in the region surrounding *SYK* (*SYK* indicated in red). A graphical representation of interactions from the 4C viewpoint (highlighted by a yellow strip) located 27 kb downstream of the *SYK* promoter is drawn with arcs at the bottom and highlighted by blue stripes. H−L refers to subtraction of the ChIP-seq signal (NSD2 High−Low). **d** Hi-C plots of the region surrounding *SYK* in NSD2 Low and High cells (left and right panels, respectively)

1 mM EDTA pH 8.0 NaOH, 0.5 mM EGTA pH 8.0 NaOH, 0.1% sodium deoxycholate, 0.5% N-lauroysarcosine). Lysates were sonicated using Bioruptor (Diagenode) (15 cycles 30 s ON, 30 s OFF, an agarose gel was run to make sure that the sonicated DNA smear was in the range of 100–700 bp).Triton X-100 1% final were added and the samples were centrifuged 5 min at 16,000 rcf at 4 °C. Supernatant

was collected. Antibody was combined with protein A magnetic beads for one hour at room temperature and added to chromatin. For CTCF, H3K27ac and Rad21 and IgG as negative control, 10 μl of antibody (Millipore 07-729, Abcam ab4729, Abcam ab992, Abcam ab37415, respectively) was added to 50 μl of protein-A magnetic beads (Dynabeads) and added to the sonicated chromatin from

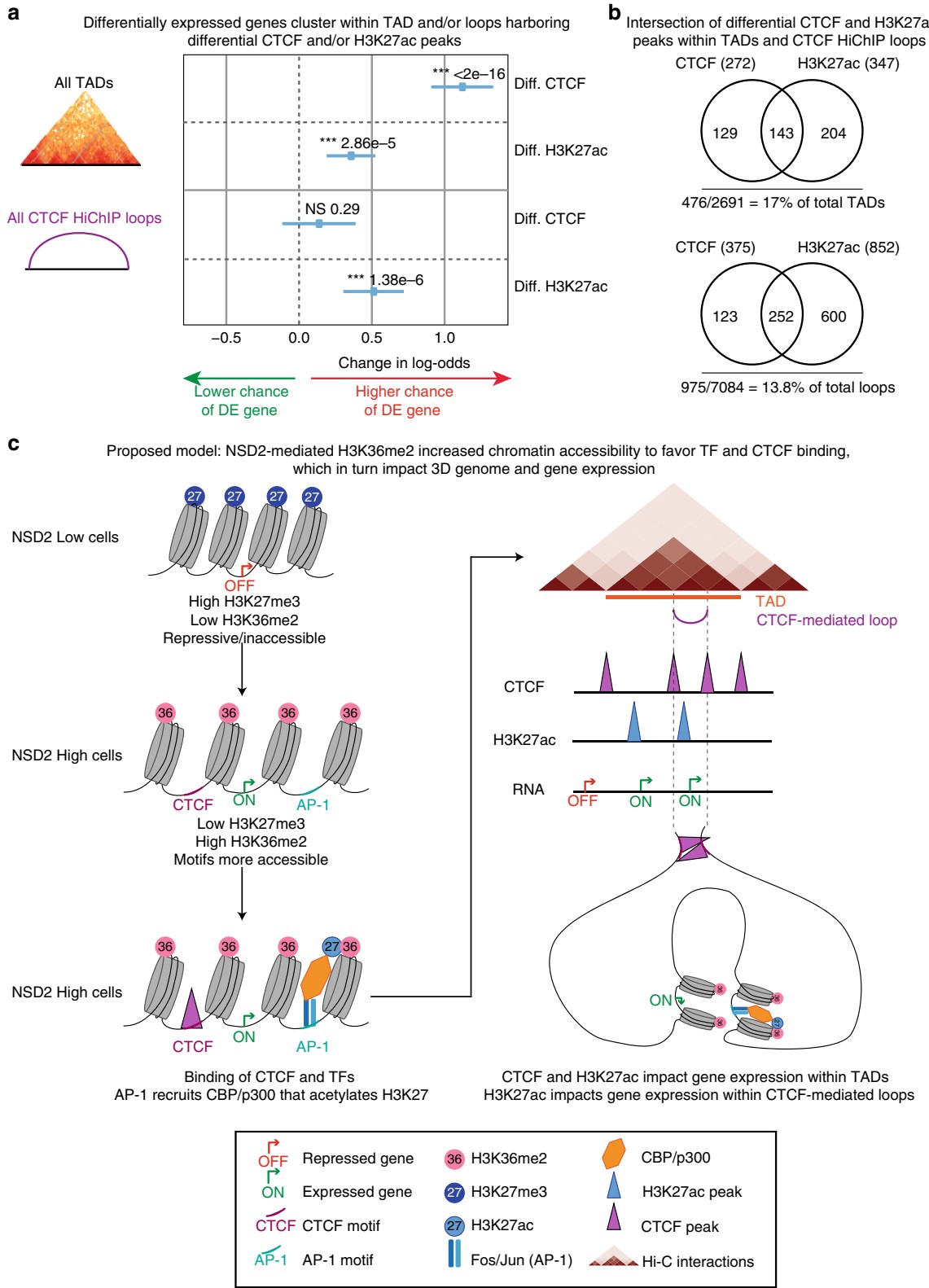

**Fig. 7** Changes in gene expression occur as a function of changes in chromatin in TADs and CTCF loops. **a** Logistic regression model of gene expression changes as a function of CTCF and/or H3K27ac changes in TADs and CTCF loops (see the "Methods" section for details). Diff. CTCF and H3K27ac: differential CTCF and H3K27ac peaks. DE gene: differentially expressed gene. *P*-values were calculated using Wald-test (***P < 2e−16 for differential CTCF in TADs; ***P = 2.6e−5 for Diff. H3K27ac in TADs; NSP = 0.29 for Diff. CTCF in loops; ***P = 1.38e−6 for Diff. H3K27ac in loops). **b** Intersection of differential CTCF and H3K27ac peaks within TADs and CTCF loops. **c** Proposed model: NSD2-mediated H3K36me2 increases chromatin accessibility to favor TF and CTCF binding, which in turn impacts 3D genome reorganization and gene expression changes

10 million cells per immunoprecipitation. For H3K36me2, internal spike-in was added for normalization as previously described[65]. Briefly, per immunoprecipitation, 1 µl of H3K36me2 antibody and 0.1 µl of Drosophila-specific H2Av antibody were added to 10 µl of protein-A magnetic beads and added to 100 µg of human sonicated chromatin supplemented with 2 µg of drosophila-sonicated chromatin. Of note, chromatin was quantified with Nanodrop at 260 nm. Immunoprecipitation was performed for 3–6 h rotating in the cold room, then washes and tagmentation were performed as per the original ChIPmentation protocol[64]. Briefly, beads were washed twice with 500 µl cold low-salt wash buffer (20 mM Tris–HCl pH 7.5, 150 mM NaCl, 2 mM EDTA pH 8.0 NaOH, 0.1% SDS, 1% triton X-100), twice with 500 µl cold LiCl-containing wash buffer (10 mM Tris–HCl pH 8.0, 250 mM LiCl, 1 mM EDTA pH 8.0 NaOH, 1% triton X-100, 0.7% sodium deoxycholate), and twice with 500 µl cold 10 mM cold Tris–Cl, pH 8.0, to remove detergent, salts and EDTA. Subsequently, beads were resuspended in 25 µl of the freshly prepared tagmentation reaction buffer (10 mM Tris–HCl, pH 8.0, 5 mM MgCl₂, 10% dimethylformamide) and 1 µl Tagment DNA Enzyme from the Nextera DNA Sample Prep Kit (Illumina) and incubated at 37 °C for 1 min in a thermocycler. Following tagmentation, the beads were washed twice with 500 µl cold low-salt wash buffer (20 mM Tris–HCl pH 7.5, 150 mM NaCl, 2 mM EDTA pH 8.0 NaOH, 0.1% SDS, 1% triton X-100), and twice with 500 µl cold Tris–EDTA–tween buffer (0.2% tween, 10 mM Tris–HCl pH 8.0, 1 mM EDTA pH 8.0). Chromatin was eluted and decrosslinked by adding 70 µl of freshly prepared elution buffer (0.5% SDS, 300 mM NaCl, 5 mM EDTA pH 8.0, 10 mM Tris–HCl pH 8.0) and 2 µl of proteinase K at 1 mg/ml for 2 h at 55 °C and overnight incubation at 65 °C. Supernatant was kept and to recover as much DNA as possible, beads were washed with an additional 30 µl of elution buffer and combined supernatant was incubated an additional hour at 55 °C. DNA was purified on a column with the Qiagen Mini Elute kit. Purified DNA (20 µl) was combined with 2.5 µl of each primer at 25 mM and 25 µl of NEB Next PCR master mix and was amplified as per the ChIPmentation protocol[64] in a thermomixer with the following program: 72 °C for 5 min; 98 °C for 30 s; 14 cycles of 98 °C for 10 s, 63 °C for 30 s and 72 °C 30 s; and a final elongation at 72 °C for 1 min. DNA was purified using two consecutive rounds of SPRI AMPure XP beads: the first one with a beads-to-sample ratio of 0.6:1 to remove potential fragments larger than 700 bp (supernatant kept) and the second one with a beads-to-sample ratio of 1:1 to remove potential primer dimers (beads kept), and eluted in 20 µl of H₂O. Samples were quantified using the Tapestation bioanalyzer (Agilent) and KAPA Library Quantification Kit and sequenced with Illumina Hi-Seq 2500 using 50 cycles paired-end mode (CTCF and H3K27ac) or single-end mode (Rad21 and H3K36me2).

**Hi-C experimental procedures.** Hi-C was performed in duplicate (two NTKO replicates from independent cultures for NSD2 High, and one replicate from each of two TKO clones for NSD2 Low), from 0.5 to 1 million cells fixed in culture medium within 1% formaldehyde at RT for 10 min and quenched with 0.125 M glycine. Hi-C samples were processed using the Arima Hi-C kit as per the kit protocol, and sequenced with Illumina Hi-Seq 2500 using 50 cycles paired-end mode.

**CTCF HiChIP experimental procedures.** HiChIP was performed in triplicate (for NSD2 High: one KMS11 and two NTKO replicates from independent cultures, for NSD2 Low, two replicates from one clone and one replicate from the other one, from independent cultures). Cells were fixed in culture medium with 1% formaldehyde at RT for 10 min and quenched with 0.125 M glycine. Pellets were washed twice with ice-cold PBS, snap-frozen in liquid nitrogen and stored at −80 °C. HiChIP was performed with 15 million cells, as per the original protocol[54]. Cells were then lysed in 500 µl ice-cold lysis buffer (10 mM Tris–HCl pH 8.0, 10 mM NaCl, 0.2% Igepal CA-630, protease inhibitor cocktail (Roche complete, EDTA-free)) rotating at 4 °C for 30 min. Cell pellets were collected, washed once in 500 µl ice-cold lysis buffer and then incubated in 100 µl 0.5% SDS at 62 °C for 10 min. SDS was quenched by adding 285 µl of H₂O and 50 µl of Triton X-100 10%, and incubating at 37 °C for 15 min. Chromatin was then digested by adding 50 µl of NEBuffer 2 10X and 350 units of MboI (NEB R0147M) at 37 °C for 2 h while rotating at 950 rpm. MboI was inactivated by incubating the samples 20 min at 62 °C. To fill in the restriction fragment overhangs and mark the DNA ends with biotin, 1.5 µl 10 mM dCTP, 1.5 µl 10 mM dGTP, 1.5 µl 10 mM dTTP, 37.5 µl 0.4 mM biotin-14-dATP (Life Technologies 19524-016), and 50 units Klenow (DNA polymerase I large fragment, NEB M0210L) were added to each tube, and incubated for 60 min at 37 °C. Ligation mix was added to the samples (150 µl 10x ligation buffer (NEB B0202S), 7.5 µl 20mg per ml BSA (NEB B9001S), 150 µl Triton X-100 10%, 4000 units T4 DNA ligase (NEB M0202S), and 655.5 µl H₂O) for 4 h at RT with rotation. Following ligation, nuclei were pelleted and resuspended in 350 µl cold Nuclei Lysis Buffer (50 mM Tris–HCl pH 7.5, 10 mM EDTA, 1% SDS, and 1x Protease Inhibitors) with incubation rotating in the cold room for 10 min. Samples were sonicated on the bioruptor for 15 min (an agarose gel was performed to make sure that the sonicated DNA smear was in the range of 250–600 bp), supplemented with 1% Triton X-100 and centrifuged 5 min at 16,000 rcf at 4 °C. CTCF antibody (5 µl, Millipore 07-729) was combined with 50 µl of protein-A magnetic beads (Dynabeads) for one hour at room temperature, and added to sonicated chromatin from 15 million cells. Immunoprecipitation was performed by overnight incubation rotating in cold room and washes were performed. Briefly,

beads were washed twice with 500 µl cold Low-salt wash buffer (20 mM Tris–HCl pH 7.5, 150 mM NaCl, 2 mM EDTA pH 8.0 NaOH, 0.1% SDS, 1% triton X-100), twice with 500 µl cold high-salt wash buffer (20 mM Tris–HCl pH 7.5, 500 mM NaCl, 2 mM EDTA pH 8.0 NaOH, 0.1% SDS, 1% triton X-100), and twice with 500 µl cold LiCl-containing wash buffer (10 mM Tris–HCl pH 8.0, 250 mM LiCl, 1 mM EDTA pH 8.0 NaOH, 1% NP-40, 1% sodium deoxycholate). Chromatin was eluted and decrosslinked by adding 100 µl of freshly prepared elution buffer (0.5% SDS, 300 mM NaCl, 5 mM EDTA pH 8.0, 10 mM Tris–HCl pH 8.0) and 10 µl of proteinase K at 10 mg/ml for 45 min at 55 °C and at least 1.5 h at 67 °C. DNA was purified on kept a column with the Qiagen mini Elute kit, eluted in 12 µl of H₂O, and quantified using Qubit. Of note, we obtained between 3 and 8 ng of DNA for CTCF HiChIP from 15 million cells. To enrich for ligation events, 5 µl of Streptavidin C-1 beads were washed in Tween Wash Buffer (TWB, 5 mM Tris–HCl pH 7.5, 0.5 mM EDTA pH 8.0, 1M NaCl, 0.05% Tween-20), resuspended in 10 µl of 2X biotin binding buffer (10 mM Tris–HCl pH 7.5, 1 mM EDTA pH 8.0, 2 M NaCl), added to 10 µl of the samples and incubated at room temperature for 15 min with rotation. Samples were then washed twice in TWB with 2 min incubation at 55 °C shaking. For tagmentation, beads were washed twice in 100 µl of freshly prepared tagmentation reaction buffer (10 mM Tris–HCl, pH 8.0, 5 mM MgCl2, 10% dimethylformamide), and resuspended in 25 µl of the tagmentation reaction buffer and 1 µl Tagment DNA Enzyme from the Nextera DNA Sample Prep Kit (Illumina) and incubated at 55 °C for 10 min in a thermocycler with interval shaking. Beads were resuspended in 50 mM EDTA and incubated at 50 °C for 30 min to quench the transposase reaction. This was followed by two washes in 50 mM EDTA incubated at 50 °C for 3 min, two washes in Tween Wash buffer incubated at 55 °C for 2 min, and one wash in 10 mM Tris-HCl pH 7.5. Beads were resuspended in 50 µl of PCR master mix (1 µl of each primer at 25 mM, 25 µl of NEB Next PCR master mix and 23 µl of H₂O) and DNA was amplified in a thermomixer with the following program: 72 °C for 5 min; 98 °C for 30 s; 10 cycles of 98 °C for 10 s, 63 °C for 30 s and 72 °C for 1 min. DNA was purified using two consecutive rounds of SPRI AMPure XP beads: the first one with a beads-to-sample ratio of 0.6:1 to remove potential fragments larger than 700 bp (supernatant kept) and the second one with a beads-to-sample ratio of 0.18:1 to keep fragments greater than 300 bp (on beads), and eluted in 15 µl of H₂O. Samples were quantified using the Tapestation bioanalyzer (Agilent) and the KAPA Library Quantification Kit and sequenced with Illumina Hi-Seq 2500 using 50 cycles paired-end mode.

**4C-seq experimental procedures.** 4C-seq was performed in duplicate (two NTKO replicates from independent cultures for NSD2 High, and one replicate from each TKO clone for NSD2 Low), and analyzed as previously described[66]. 10 million cells were fixed in 2% formaldehyde for 10 min at room temperature and quenched with glycine (0.125 M final concentration). Nuclei were isolated in lysis buffer (50 mM Tris–HCl pH 7.5, 150 mM NaCl, 5 mM EDTA, 0.5% NP-40, 1% TX-100 containing 1X Roche complete mini protease inhibitors) and dounced 40 times on ice. Nuclei were resuspended in 360 µl H₂O and 60 µl warm 10X NEB DpnII buffer. They were permeabilized using 15 µl 10% SDS for 60 min at 37 °C and then 150 µl 10% Triton X-100 for 60 min at 37 °C. Chromatin was digested using 500U DpnII (NEB) overnight at 37 °C while shaking, and the digestion was repeated with an additional 250U of enzymes for 8 h, meanwhile digestion was determined by gel electrophoresis. Enzyme was deactivated at 65 °C for 20 min. Chromatin samples were divided in three tubes, diluted and ligated by adding H₂O up to 1.2 ml, 133 µl T4 ligase buffer 10X and 6000U total NEB T4 DNA Ligase (M0202M) per tube and incubating at 16 °C overnight while shaking. Ligation efficiency was checked by gel electrophoresis. Chromatin was de-crosslinked with proteinase K overnight at 65 °C, and treated with RNase A at 37 °C for 1 h. DNA was extracted by Phenol:Choroform and precipitated with Ethanol. Purified DNA was digested with 50U NEB Csp6I overnight at 37 °C while shaking, and digestion was determined by gel electrophoresis. Enzyme was deactivated at 65 °C for 20 min. DNA circularization was performed using 4000U NEB T4 DNA Ligase overnight at 16 °C. A total of 1 µg DNA was amplified per sample with inverse PCR primers containing Illumina forward and reverse sequencing adapters (see Key resources for sequences). PCR was performed using Expand™ Long Template PCR System (Sigma) with the following thermocycler program: 94 °C for 2 min; 94 °C for 15 s; 53–55 °C for 1 min; 68 °C for 2.30 min; repeat for 29 cycles; 68 °C for 7 min; hold at 4 °C. 4C-seq libraries were size-selected on gel to remove any potential primer dimers and fragments above 700 bp, then quantified using RT-PCR (KAPA Biosystems) and sequenced using 50bp single-end on Illumina HiSeq 2500.

SYK 4C bait primers:

DpnII restriction site:

AATGATACGGCGACCACCGAGATCTACACTCTTTCCCTACACGACGC
TCTTCCGATCTNNNNNNGAGGGCATTCCCATTAGATC (NNNNNNN: barcode sequence different for each sample).

Csp6I restriction site:

CAAGCAGAAGACGGCATACGAGATAGGTCGCAGTGACTGGAGTTCAG
ACGTGTGCTCTTCCGATCTtaatctttggataagtggcc.

**RNA-Seq data processing and quality control.** Paired-end reads were mapped to the hg38 genome using TopHat2[67] (parameters:–no-coverage-search–no-discordant–no-mixed–b2-very-sensitive–N 1). Bigwigs were obtained for visualization on individual as well as merged bam files using Deeptools/2.3.3[68] (parameters

bamCoverage–binSize 1–normalizeUsing RPKM). Counts for Refseq genes were obtained using htseq-counts[69]. PCA was performed using R to check the reproducibility of replicates (See Supplementary Fig. 1a). DESeq2 version 1.4[70] was used to remove genes for which there are less than 3 samples with counts greater than or equal to 5 (13,603 genes remained), normalize expression counts and get differentially expressed genes (absolute log2-fold-change > 1 and FDR < 0.01 using the Wald test). Gene Set Enrichment Analysis[71] for pathway enrichment was performed using the default parameters of the GSEA desktop application on normalized reads counts for each NSD2 High and Low RNA-seq replicate from the 26,586 protein coding genes of hg38 genome. Gene expression changes are associated with H3K27ac and CTCF changes at promoters (top panel, −/+3kb around TSS), distal H3K27ac and CTCF and super-enhancers (middle and bottom panels, respectively, 3–250 kb up and downstream from TSSs associated with genes using GREAT).

**ATAC-seq data processing and quality control**. Reads were aligned to hg38 genome with Bowtie2[72] (parameters: –no-discordant -p 12 –no-mixed -N 1 -X 2000). Potential PCR duplicates were removed from the reads with Picard-tools. ATAC-seq peaks were called with PeaKDEck[73] (parameters: -sig 0.0001 -PVAL ON). Bigwigs were obtained for visualization on individual as well as merged bam files using Deeptools/2.3.3[68] (parameters bamCoverage–binSize 1–normalizeUsing RPKM). transcription factor affinity prediction (TRAP[74]) was used to identify which motifs were present in the ATAC-seq peaks.

**4C-seq data processing and quality control**. Processing of 4C-seq data was performed using 4Cker pipeline[66]. Briefly, mapping was performed using Bowtie2[72] to a reduced genome consisting of all unique 24-nt-long regions surrounding DpnII sites from the human reference genome (hg38), allowing for zero mismatches. For comparison between conditions, DESeq2 version 1.4[70] with default parameters was used to normalize total read count per window between samples and to identify the windows with significant 4C signal differences, using an FDR cutoff of 0.01.

**ChIP-seq data processing and quality control**. Reads from our ChIP-seq as well as publically available H3K27me3 ChIP-seq (GSE57977) were aligned to hg38 genome with Bowtie2[72] (parameters: –no-discordant -p 12 –no-mixed -N 1 -X 2000). Ambiguous reads were filtered to use uniquely mapped reads in the downstream analysis. PCR duplicates were removed using Picard-tools (version 1.88). Bigwigs were obtained for visualization on individual as well as merged bam files using Deeptools/2.3.3 (parameters bamCoverage–binSize 1–normalizeUsing RPKM; or bamCompare–verbose–binSize 25–ratio subtract–scaleFactorsMethod SES for subtraction files). For H3K36me2 ChIP-seq, an internal Drosophila spike-in was added. Reads were aligned to dm6 genome with Bowtie2 (parameters: –no-discordant -p 12 –no-mixed -N 1 -X 2000). Bigwigs were created after normalization with the spike-in Drosophila read counts. Heatmaps and average profiles were performed on merged bigwig files using Deeptools/2.3.3. For CTCF, H3K27ac and Rad21 ChIP-seq, MACS version 1.4.2[75] was used to call peaks (parameters: -p 1e-6 -g hs -B–single-profile). For CTCF and H3K27ac ChIP-seq, a reference list of peaks coordinates was created containing all the peaks present in any replicate, and merging overlapping peaks using Bedtools merge –i. Counts for the reference list of peak coordinates were obtained using htseq-counts[69]. PCA was performed using R to check the reproducibility of replicates (see Supplementary Figs. 2a and 3a). DESeq2 version 1.4[70] was used to normalize read counts and identify differential peaks (absolute log2-fold-change > 1 and FDR < 0.01 using the Wald test) and differential super-enhancers (FDR < 0.1 using the Wald test). Overlap of super-enhancers (Supplementary Fig. 2b) was performed using Bedtools with the following parameters: -f .25 -u -r –wa. For CTCF ChIP-seq, the MEME-ChIP tool from the MEME suite was used to detect CTCF motifs in increased peaks and equivalent number of peaks randomly picked from the full list of peaks. Genomic Regions Enrichment of Annotations Tool (GREAT[38]) was used to associate peaks at distal sites and super-enhancers with gene expression changes, using an arbitrary cutoff of 250 kb, based on findings from a previous study showing that the mean distance from enhancer to promoter is around 196 kb[39].

**Statistical analysis**. Supplementary Table 2 indicates the tool used for each statistic test, the cutoff applied and whether default parameters were used.

**Hi-C Data processing**. HiC-Bench[50] was used to align and filter the Hi-C data, identify TADs, and generate Hi-C heatmaps. To generate Hi-C-filtered contact matrices, the Hi-C reads were aligned against the human reference genome (hg38) by bowtie2[72] (version 2.3.1). Mapped read pairs were filtered by the GenomicTools[76] tools-hic filter command integrated in HiC-bench for known artifacts of the Hi-C protocol. The filtered reads include multi-mapped reads ('multihit'), read-pairs with only one mappable read ('single sided'), duplicated read-pairs ('ds.duplicate'), low mapping quality reads (MAPQ < 30), read-pairs resulting from self-ligated fragments, and short-range interactions resulting from read-pairs aligning within 25kb ('ds.filtered'). For the downstream analyses, all the accepted intra-chromosomal read-pairs ('ds.accepted intra') were used. The Hi-C filtered contact matrices were corrected using the ICE 'correction' algorithm built into

HiC-bench. TADs and boundaries were identified using the Crane[77] method at 40 kb bin resolution with an insulating window of 500 kb. HiC heatmaps for regions of interest were generated using the ICE-corrected contact matrices through the 'hic-plotter-diff' pipeline step integrated in HiC-bench.

**Hi-C data quality control and TAD/Boundary Stats**. Quality assessment analysis shows that the total numbers of reads in the samples ranged from ~120 million to ~185 million (Supplementary Fig. 6a). The percentage of reads aligned was always over 98% in all samples. The proportion of accepted reads ('ds-accepted-intra' and 'ds-accepted-inter') were in the range of ~46–47%.

The number of TADs identified by the Crane method across replicates ranged between 3460 and 3532 from which ~500 TADs (14%) showed sizes smaller or equal to 80 kb in each replicate. We removed such short length TADs from the following metrics analysis, considering this approach more representative. The distribution of TAD sizes showed that 85% of the TADs ranged between 160 kb and 1.04 Mb with a median of 480 kb in both NSD2 High and Low conditions. The mean TAD sizes were 573 and 591 kb for NSD2 High and Low conditions, respectively (Supplementary Fig. 6b).

R (prcomp, scale = TRUE and center = TRUE) was used to perform a PCA on the Hi-C datasets using the 'ratio' insulation scores produced by HiC-Bench (bins = 40kb) after ICE correction (Supplementary Fig. 5b).

**Screening of potentially altered TAD boundaries**. The Hi-C downstream analysis involved a genome-wide screening of TAD boundary insulation changes in NSD2 High cells based on boundary insulation scores (ratio index) and CTCF/Rad21-binding enrichment. Individual cases of potentially altered boundaries were confirmed by visual inspection of HiC heatmaps.

**Mean Boundary Insulation Scores (Ratio index)**. To assess and compare boundary strength alteration in NSD2 High versus Low cells, we included the calculation of the Mean Boundary Score (MBS) for every boundary identified in the NSD2 Low condition (reference boundaries). For this purpose, we used the 'by group' NSD2 Low TADs identified by the Crane method. HiC-Bench performs a 'by group' analysis by merging the DNA interaction information of all the replicates of the same condition, in this case to identify TADs per condition.

We calculated the MBS as the arithmetic mean of the 'ratio' insulation scores inside the reference boundary coordinates being assessed. HiC-Bench calculates one ratio score per bin (40 kb) as explained in Lazaris et al.[50]. As a result, there are generally multiple insulation scores per TAD boundary identified by the Crane algorithm. The number of bins inside boundaries showed a median number of 7 and, accordingly, the boundary median size was 280 kb. The MBS was used to calculate NSD2 High MBS logFC values with respect to the NSD2 Low condition. A differential analysis on the ratio insulation scores inside each boundary was also performed. An unpaired t-test (two-sided) was used by pooling all the ratio insulation scores inside the reference boundary coordinates and adjusting with FDR correction.

**CTCF and Rad21 occupancy in boundaries**. The CTCF and Rad21 peaks were mapped to the boundaries to integrate the boundary insulation data obtained with the CTCF and Rad21-binding data. We assigned a peak to a boundary if the peak overlapped with the boundary region (>0 bp). An extension of the boundary region by 1 bin (40 kb) on either side of the boundary was considered. The CTCF/Rad21 signal of all the peaks assigned to a boundary were aggregated and then NSD2 High versus Low logFC values were calculated. Significant changes in global CTCF/Rad21 occupancy within the boundaries were calculated using a two-sided unpaired t-test by pooling all the CTCF/Rad21 peak intensities assigned in the boundary.

A ranked-table of boundary coordinates with the insulation and CTCF/Rad21 metrics was created. To generate Hi-C heatmaps, the best-ranked boundary cases were selected by taking into account unidirectional significant fold changes in MBS, CTCF and Rad21 (FDR < 0.01). In this approach, we assumed that there would be a positive correlation between boundary insulation scores and CTCF/Rad21 binding in boundaries. In addition, we screened each boundary candidate and the adjacent TADs and boundaries in search of significant deregulated genes, gain/loss of interactions and significant changes in H3K27ac.

**Compartments analysis**. We grouped the cis-eigenvector1 values (50kb bins) in 50 equally sized ranks (pooled replicates) sorted in ascending compartment signal values (strongest B-compartment like bins to strongly A-compartment like bins) and computed pairwise log2 enrichment of contact counts (IC-normalized) between all groups (from left to right and from top to bottom: the 50 groups of eigenvalues bins ranked from strongest B-like to strongest A-like bins). Compartmentalization was measured as explained in Schwarzer et al.[9]: $\sqrt[2]{AA * BB / AB^2}$. We also calculated compartmentalization strength as in Nora et al.[8]: $\frac{median(AA \cup BB)}{median(AB \cup BA)}$, with AA, BB, AB, and BA in the strongest 20% (Supplementary Fig. 5c).

Analysis of switching compartments regions was carried out using the Homer pipeline (v4.6)[77]. Homer performs a PCA of the normalized interaction matrices and uses the PC1 component to define regions of active (A compartments) and

inactive chromatin (B compartments). HiC filtered matrices were given as input to run Homer with default parameters (50 kb resolution). To confirm the proper sign of the A and B compartment, we used the–active parameter to input peaks of the active mark H3K27ac. We use Homer to compare the interaction profiles in both experiments and calculate a correlation. If one region interacts similarly with other regions in both conditions then the correlation will be high. On the other hand, the correlation will be low if a locus interacts with different regions in both conditions. Using Homer's getHiCcorrDiff.pl, we compared the interaction profiles of both conditions and obtained a correlation difference to identify stable and switching compartments. Altered compartments are named as AB and BA for regions switching from A to B and B to A, respectively.

**Integration of AB and BA compartment with others datasets**. To show the correlation between the different measurements and the compartment changes, we mapped the peaks obtained in H3K27ac2, CTCF, RAD21 and ATAC-seq to the AB, BA, and Stable regions. We used the peak intensity values to calculate the peak intensity fold change between NSD2 High and Low and the mean fold change of all the peaks assigned to a compartment region. Similarly, in the case of RNA-seq data, genes were mapped to the compartment regions and the mean fold change of all the genes assigned to a region was calculated by using the DESeq2 fold change data.

We assigned a peak or gene to a compartment region when the complete peak or gene coordinate was found inside the compartment coordinates. For sparse measurements as in H3K36me2 and H3K27me3 chromatin marks, we used HTSeq to obtain read counts on the AB, BA, and Stable regions. Next, we used DESeq2 to normalize the read counts across NSD2 High and Low replicates and to obtain fold change values.

To show the correlation between the different measurements and the compartment changes, we used the mean fold change values to generate boxplots. Statistical significance was assessed using a paired two-sided Wilcoxon rank-sum test.

**Intra-TAD interactions**. To assess statistically significant intra-TAD interactions, we used an algorithm developed by Kloetgen et al.[78]. As a first step, the algorithm identifies overlapped or positionally consistent TADs (common TADs). This approach establishes a minimum TAD length of 10 bins (400 kb) and extends either side of the TAD by three bins (±120 kb in 40 kb resolution). TADs across two samples are considered positionally consistent if their boundaries are as close as three bins. The boundaries of the common TAD are then set to those that yield the largest TAD. The set of common TADs between any two samples $s_1$ and $s_2$ is denoted as T. In the next step, a paired two-sided $t$-test is performed on each single interaction bin within each common TAD between the two samples. It calculates the difference between the average scores of all interaction intensities within such TADs. A multiple testing correction by calculating the false-discovery rate per common TAD (using the R function p.adjust with method = 'fdr') is also calculated.

$$\text{TAD interactions change}(t) = \left( \frac{\sum_{i \in I_t} S_{2_i}}{\#I_t} \right) - \left( \frac{\sum_{i \in I_t} S_{1_i}}{\#I_t} \right) \qquad (1)$$

for each $t \in T$, and $I_t$ being all intra-TAD interactions for TAD $t$. We classified the common TADs in terms of Loss, Gain or Stable intra-TAD interactions by using FDR < 0.1 and absolute TAD interactions change >0.3.

**Identification of Gain, Loss and Stable TADs**. We used the same method as described in the previous subsection to integrate the H3K27ac2, CTCF, RAD21, ATAC-seq, RNA-seq data, H3K36me2, H3K27me3 measurements with the different intra-TAD interactions subgroups described above (Gain, Loss, and Stable TADs).

**CTCF binding, H3K27ac, and RNA expression in common TADs**. To compute the total number of differential changes in CTCF binding, RNA expression, and H3K27ac mark that fall within common TADs. We overlapped all CTCF peaks, genes, and H3K27ac peaks with common TADs. Volcano plots were generated using the log2-fold changes and –log10 (p-value) for all three data types (CTCF-ChIP, RNA-seq, H3K27ac-ChIP). Overlapping features with log2-fold change >1 and a q-value <0.01 were colored red, while overlapping features with log2-fold change less −1 and a q-value <0.01 were colored in blue, and all others were colored in black.

To assess the correlation of CTCF binding, H3K27ac, and RNA expression inside each Common TAD, we assigned a peak or gene to each Common TAD when the complete peak or gene coordinate was found inside the TAD coordinates. Common TADs lacking either CTCF, H3K27ac, or RNA expression features were filtered. The mean fold change of each feature (CTCF, H3K27ac, and RNA expression) inside each Common TAD was computed by two methods. One method considered only the differential peaks/genes found inside a Common TAD (FDR < 0.05) in the mean fold change calculation. The second method considered all the peaks/genes assigned to a Common TAD.

**Intra-TAD interactions and compartment alteration**. One mean log fold change value for each feature (CTCF, H3K27ac and RNA expression) was assigned to every Common TAD, together with the intra-TAD interaction fold change value previously calculated (see 'Intra-TAD interactions' subsection). Compartment alteration was also assessed by calculating the mean PC1 value of each Common TAD (50 kb bins) and computing the PC1 mean difference between conditions (NSD2 High–NSD2 low). A positive value of the PC1 mean difference indicates that in the NSD2 High condition, that Common TAD had become more active. The higher the PC1 mean difference the stronger the compartment alteration change. To confirm if the PC1 mean difference correlated with real compartment changes, we looked at whether the significant AB and BA regions identified by Homer overlapped with the Common TADs. We then computed the overlap length.

To show the intra-TAD association of all the five features (CTCF, H3K27ac, RNA expression, Intra-TAD interactions and PC1 mean difference) we classified the Common TADs in 'positive', 'negative' or 'no correlated' groups, by taking into account the direction of all the five features (positive correlation). The Pearson correlation coefficient (PCC) was calculated using R ('Cor' and 'pairs' command). 3D plots were also generated using R ('plot_ly' command of the 'plotly' library).

**HiChIP data processing and quality control**. HiChIP paired-end reads were aligned to hg38 genome using the HiC-Pro pipeline. Default settings were used to remove duplicate reads, assign reads to MboI restriction fragments, filter for valid pairs, and generate 10 kb binned interaction matrices. FitHiChIP[55] was used to identify statistically significant chromosomal interactions from the CTCF HiChIP experiments, using a bin size of 10 kB. FitHiChIP allows users to either infer peaks from their HiChIP data or use a reference list of peaks from ChIP-seq data. Statistically significant HiChIP interactions (q < 0.01) were called separately in each experiment with a minimum distance of 20 kB and a maximum distance of 3 MB. The UseP2PBackgrnd parameter was set to 1 for peak to peak background estimation. HiChIP interactions were considered statistically significant if a peak was found in at least one anchor (Peak to all interactions). This resulted in 8651 statistically significant CTCF peak to all interactions in NSD2 High and 4914 CTCF peak to all interactions in NSD2 low. Common CTCF high confidence loops in NSD2 high and NSD2 low were identified if both pairs of loop anchors are overlapping. The median size for common loops was calculated to be 210 kB. Boxplots were generated to show the mean contact counts for common CTCF loops in NSD2 low and NSD2 high experiments.

**HICHIP integration with RNA-seq and CTCF/H3K27ac ChIP-seq**. To compute the total number of differential changes in CTCF binding, RNA expression, and H3K27ac marks that fall within common CTCF HiChIP loops, we overlapped all CTCF peaks, genes (5 kb upstream of TSS), and H3K27ac peaks within common HiChIP loops (full loop plus anchors). Volcano plots were generated using the log2-fold changes and –log10 (p-value) for all three data types (CTCF-ChIP, RNA-seq, H3K27ac-ChIP). Overlapping features with log2-fold change >1 and a q-value <0.01 were colored red, while overlapping features with log2-fold change less −1 and a q-value <0.01 were colored in blue, and all others were colored in black.

To assess the correlation of CTCF binding, H3K27ac, and RNA expression inside each Common CTCF Loops, we applied the same approach used to assess the correlation of these features in Common TADs. First, all CTCF peaks were overlapped with all the common loops, and the mean log2-fold change for all peaks found within a common loop were calculated. The same approach was applied to calculate the mean log2-fold change for all genes and for all H3K27ac peaks within each common loop.

To correlate the mean log2-fold changes of all three features, we considered only the common loops that had at least one overlapping differential CTCF and H3K27ac peak, as well as one gene expression change. Pairwise correlation plots were generated for the three comparisons, so that each individual point in the scatterplot represents a common CTCF loop. Loops that had positive mean log2-fold change for all three features were colored in red, while loops that had negative log2-fold change for all three features were colored in blue.

To determine if there is a correlation between the significant alterations of CTCF, RNA expression, and H3K27ac, we performed the same analysis, but first filtered for significant changes in CTCF, RNA expression, and H3K27ac peaks, with a q-value of <0.05 and an absolute value of the log2-fold change >1. Next, the mean log2-fold change was computed for all the differential CTCF, gene expression changes, and H3K27ac peaks that overlapped common HiChIP loops. The Pearson correlation was calculated for all three comparisons.

**Logistic regression model**. To evaluate which factors contribute most to gene regulation, we modeled the probability that a gene is differentially expressed as a function of chromatin features in insulated domains (TADs and/or CTCF loops) using logistic regression. The covariates of our model were four binary variables which indicate whether each gene shares a CTCF HiChIP loop or a TAD with at least one differential CTCF or H3K27ac peak (data.frame = Supplementary Data 4). The model was fitted using R's glm function as following:

glm(gene.de ~k27ac.tad + ctcf.tad + k27ac.loop + ctcf.loop, family = binomial(), data = data.frame). The standard Wald-test was used to access the significance of each factor.

## Data availability

Data are available at NCBI GEO under the accession number GSE131651. All other relevant data supporting the key findings of this study are available within the article and its Supplementary Information files or from the corresponding author upon reasonable request. The source data underlying Figs. 1g, 2c, 3e, 4b, d, 5b, c and Supplementary Figs. 1c, 5d, 6c and 9a, b are provided as a Source Data file. A reporting summary for this Article is available as a Supplementary Information file.

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

## Acknowledgements

The authors thank the Ben Ho Park laboratory for sending the NSD2 High and Low cells, Skok lab members for helpful scientific discussions, New York University School of Medicine High Performance Computing (HPC) for computing technical support, and Adriana Heguy and the Genome Technology Center (GTC) core for sequencing efforts. This work was supported by 1R35GM122515 (J.A.S.), AACR Takeda Multiple Myeloma fellowship (P.L.), National Cancer Center (P.L.), American Cancer Society (RSG-15-189-01-RMC) (A.T.) and St. Baldrick's foundation (581357) (A.T.).

## Author contributions

Conceptualization & study design, P.L. and J.A.S.; Investigation, P.L.; Formal analysis, P.L., Sa.Ba, J.-R.H., T.S., G.S., A.K., M.C., So.Bh.; Writing—original draft, P.L. and J.A.S.; Supervision, R.B., F.A., A.T. and J.S.; Funding acquisition, P.L., J.S., A.T.

## Competing interests

The authors declare no competing interests.
