## [Peer Review File · Nature Communications]

Reviewers' comments:

Reviewer #1 (Remarks to the Author):

An interesting and comprehensive paper studying the epigenomic changes including the changes in enhancers, CTCF binding and chromatin looping that occur in response to high levels of MMSET/NSD2 in an isogenic pair of multiple myeloma cell lines. Major conclusions are that the changes in global chromatin modifications affects CTCF binding, the architecture of superenhancers and looping. This is an appropriate publication for Nature Communications after addressing some modest questions

1. Figure 2e H3K27Ac peaks are ranked, H3K36me2 levels were higher in these peaks but this may be due to the global increase in H3K36me2. What is the H3K36 levels of a randomly selected set of sites- The statement that the increased H3K27Ac is associated with H3K36me2 may be true/true and unrelated.
2. Figure 2e there is some decrease in H3K27me3 in the most noticeable H3K27Ac peaks, these two marks are mutually exclusive. Text on page 9 should be modified accordingly – I don't think that the increase in H3K27Ac is unrelated to H3K27me3
3. Again there is an increase in ATAC seq peaks associated with increased H3K36me2 – but this may be true at any site selected. Nevertheless it is clear that there are genes activated and more enhancers and more CTCF binding in the NSD2 high cells.
4. Figure 2f needs better description what do the red and blue track mean – I am not certain which is in the high versus low cells – figure 3 indicates that these may be subtraction tracks- please clarify this in the figure legend of figure 2. Some of the supplemental figures explicitly say that subtraction tracks or presented- please be consistent .

The rest of the paper is really a comprehensive and extensive analysis of TADs compartments (please for readers define earlier on what are A and B compartments- this is defined in the supplemental but never in the main text) showing that oncogene up regulation is associated with switching to the active compartment, new CTCF and H3K27ac sites and stronger intra-TAD interactions. It is interesting that AP1 sites are enhanced in the H3K27Ac peaks, are there other motifs enriched as well?

Reviewer #2 (Remarks to the Author):

Transparency statement: this review was written by Guillaume Filion (CRG, Barcelona). I accepted the assignment on May 17, 2019 and I turned in my review on May 29, 2019. I worked approximately 10 hours and 30 minutes on this task. I feel competent to review the genomics methods and the statistics but I lack the biological expertise in multiple myeloma. I have reviewed the GEO records and the statistics in the manuscript, but I have not replicated them.

In the article titled "NSD2 overexpression drives clustered chromatin and transcriptional changes in a subset of insulated domains", Lhoumaud et al. investigate the dosage of the histone-methyltransferase NSD2 in a cell line model for multiple myeloma. The question is relevant because NSD2 can be up-regulated in multiple myeloma following a common translocation. Knocking out the two NSD2 alleles independently allows the authors to change the dosage of NSD2 in an isogenic context, and to observe the consequences directly.

The authors focus on gene expression, on the epigenome and on genome conformation. They observe changes at all levels, and find that they are usually consistent with the spread of the mark H3K36me2 (deposited by NSD2). For instance, H3K27ac and CTCF are typically gained in regions of heightened H3K36me2; with concomitant appearance of chromatin loops and changes in TAD borders and AB compartments.

Overall, the study is well conducted and the results are very coherent. This work is particularly relevant to tackle the question of the relationships between the epigenome and genome conformation, with a clinical relevance for cancer. In other words, I am very positive on the scientific quality of the work.

That said, I have some comments for improvements that mostly have to do with the statistics and the clarity of the text. The following issues are not all severe, but they are all important and I recommend the authors to address them the best they can.

1. There is too little information about the experimental design.

1a. Page 6, second sentence of the second paragraph: "Importantly, NTKO and TKO cells differ solely in the level of NSD2 they express and henceforth are referred to as NSD2 High and Low cells". This is absolutely crucial for the study, yet the manuscript is not explicit whether the translocation leaves the gene NSD2 completely intact. This needs to be clarified in the text and possibly in the figures.

1b. The GEO record refers to NSD2 High as wild type. This adds to the confusion and should be clarified on the record.

1c. What is meant by "NSD2 High" and "NSD2 Low" exactly? Figure 1a labels normal plasma cells and "TKO" cells as "NSD2 Low", and the KMS11 and "NTKO" cells as "NSD2 High". For instance, was the parental KMS11 used for any experiment presented in the manuscript? The supplementary material for ATAC-seq mentions something about KMS11, but later specifies that only TKO and NTKO cells were used.

The authors seem to have two "TKO" clones, so how were replicates handled? Were the two clones systematically used in every experiment as "biological" replicates? For instance, in Supplementary Figure 1a, there are 4 points labelled NSD2 Low and 4 points labelled NSD2 High. Yet the legend only says "PCA of RNAseq replicates for NSD2 High and Low cells". Does that mean two "technical" replicates of each NSD2 Low clone and four "technical" replicates of the NTKO cell? Meanwhile the RNA-seq paragraph mentions "RNA was extracted from 5 replicates for each condition". If this is so, why are there only 4 points on Figure 1a? A little further, Supplementary Figure 2a shows 3 points each and the legend simply says "PCA of H3K27ac ChIP-seq replicates for NSD2 High and Low cells". But now it is clear that there must be some imbalance in the design because there is an odd number of points.

Given the confusion about the meaning of "replicate" throughout the manuscript, I recommend the authors to update the method section to leave no doubt about this, nor about how replicates were discarded. For instance, the description of the ATAC-seq procedure is very clear (except the mention of KMS11 at the beginning), so the authors could just use this as a template for the other sections.

2. I would like to raise the following issues with the statistics.

2a. The criteria for significance keep changing throughout the manuscript. Sometimes they are based on p-values, sometimes they are based on FDR. For p-values, the cut-off for significance seems to be 0.05, but for FDR it can be 0.1, 0.05, or 0.01. I strongly recommend the authors to uniformize their approach to prevent unconscious p-hacking. At the very least they have to explain why they switch from FDR to p-values and back, and what is the motivation for using a given cut-off. My personal opinion (which I do not wish to impose) is to use FDR with a fixed cut-off for selection procedures and p-values for single tests. For instance I would have used FDR to represent the data of Figure 1g.

2b. The reports are sometimes incomplete. For instance, the statistical tests are not always

described when a p-value is provided.

2c. The reports of the FDR are sometimes awkward. For instance, one can read "... FZD8 (log₂ fold-change = 3.16, FDR = 8.18E-34)". The concept of FDR is by definition attached to a set and not to an individual (here a gene). Algorithms usually compute a certain FDR score in order to include or exclude an individual in the set, but unlike p-values, each individual FDR score depends on all the genes. Thus, considering the value in isolation is not so meaningful. It is more coherent to simply say that the gene is in a set of chosen FDR. This also avoids the problem of reporting "exotic values" such as 8.18E-34, which give a false impression of confidence.

3. Page 17, third paragraph, first sentence: "Since the NSD2 phenotype relies on the histone methyl transferase activity of the protein". I somehow missed the evidence where the authors prove this claim in their model. The rest of the paragraph is to the point, but here the authors should show the evidence that the methyltransferase activity and no other potential function of NSD2 drives the changes they observe (perhaps was it done by the creators of the cell line), or they should rephrase the conclusion.

4. As currently described, the logistic model recapitulates the previous results but adds little insight. The results of Figures 7a and 7b would be better presented as a statistical test (using Fisher's exact test for instance) because the model is used neither for inference nor for prediction. This is somewhat unfortunate because the chosen vocabulary suggests a causal relationship (e.g., "We found that changes in H3K27ac significantly increase the probability that a gene sharing the same CTCF loop is differentially expressed") whereas the discussion clearly shows that this is not the intention of the authors. A relatively easy way to improve this part is to use the model to test the interaction term between CTCF and H3K27ac in order to see whether they synergize to account for gene expression. This is typically what models are used for, so it would justify the approach. Alternatively, the authors could replace this part by a standard descriptive approach coupled to a test.

The following issues are less important scientifically, but they damage the clarity of the manuscript, so I recommend the authors to address them (most of them are fairly easy to fix).

1. Abstract: "a linear regression model" is referred to as a logistic regression model in the text. I believe the latter is the correct one, but in any case the text has to be consistent.

2. First paragraph, page 3: "1MB sized highly self-interacting 'topologically associated domains' (TADs)" is inaccurate. TADs are not all 1MB in size, so "MB sized" or "Mb size" or "Mbp sized" would be more appropriate. Also, "highly self-interacting" is subjective. Actually, polymer models suggest the opposite. For instance, the work by Luca Giorgetti et al. on the X chromosome (PMID: 24813616, figure 2G) shows that many structures are compatible with the same TAD, many of which are not particularly self-interacting. Some unpublished work by Geoffrey Fudenberg and Leonid Mirny also shows that TADs are nowhere near "isolated" as a polymer folded in 3D space. Here the authors should either stick to objective definitions or back up their statement with a citation so that we know what they mean by "highly self-interacting".

3. Page 4, 7 and 8 lines from the end: the acronym HMT is defined but never used in the manuscript. The acronym MM is defined and used only in Figure 1a. The best option is to not define the acronym HMT at all (the text reads well without it) and to define the meaning of MM in the legend of Figure 1a. Also note that "MM cell lime" should be "MM cell line" in the panel.

4. Page 5, 9 lines from the end: Rad21 appears for the first time in the manuscript, without mention that it is a cohesin component. This will be mentioned at the next mention of Rad21 in the text, so the authors should move the phrase "(a component of the cohesin complex)" here.

5. Page 6, first sentence of the second paragraph: it would be so much easier to understand if "a

patient derived" were replaced by "the patient-derived". This looks like a detail, but I actually had to run a Google search on KMS11 to make sure that I understood the sentence properly. The same comment applies to the legend of Figure 1a.

6. Page 6, last sentence of the second paragraph: "Genes associated with multiple myeloma and KRAS pathways were enriched as shown by Gene Set Enrichment Analysis (GSEA, Supplementary Fig. 1b)". This statement is not clear. In which context are those genes enriched, and relative to what? It surmise that this means that the 1953 genes that are differentially expressed contain more genes associated with myeloma and KRAS than expected from a random set of 1953 genes, but this is not clear without more context information.

7. Page 6, two lines from the end: "Since the vast majority of regulatory elements are located within intergenic regions". Is this the case? Figure 1g suggests the opposite. Here the authors could backup their statement with a citation, and refine it for consistency with their own results because they seem to distinguish intergenic regions from promoters.

8. What is the information of Figure 1d? It is mentioned in the text together with figure 1c, but no specific comment is made about it. Similarly, what is the information of Figure 2b? The authors should either comment these panels and explicitly mention why they are relevant for their scientific argument, or drop them altogether.

9 . Page 8, second paragraph, second sentence from the end: "possibly because this is the most accurate prediction". I do not understand the meaning of this statement. The authors should clarify it or remove it.

10. Page 8, 5 lines from the end: "precise transcriptional control is exerted by restricting the influence of enhancers to target genes within TADs whose boundaries are enriched for the insulating protein, CTCF". This statement could be backed up by a citation so that we know what the authors have in mind here. For instance, this is dubious in Drosophila where TADs are not really flanked by CTCF sites. Likewise, transcriptional control is sometimes achieved in different ways. Think of how viruses can tightly regulate the expression of their genes independently of enhancers. In summary, giving some context would make this statement more concrete.

11. Page 9, line 4: "Increased CTCF peaks were predominantly located in intergenic and intragenic regions, as compared with decreased and stable CTCF peaks (Fig. 2c)". I do not understand this statement. Perhaps "and intragenic" was not meant to be there?

12. Page 9, second paragraph, second sentence: the logic of this passage is somewhat confusing. "Importantly, new H3K27ac and CTCF peaks were found to be located in newly enriched H3K36me2 regions while H3K27me3 remained unchanged at these locations". The grammar of this sentence suggests that H3K27me3 does not change at the newly enriched H3K36me2 regions, but the figure shows that it does not change at the new H3K27ac and CTCF peaks. For this reason, the next sentence "Thus, increases in H3K27ac peaks were found to be independent of H3K27me3 changes" does not appear as the logical consequence of the previous one. Clarifying the first sentence would solve the issue.

13. Page 10, first sentence: "The vast majority of chromosomal loops involve cohesin". This should be backed up by a citation. There is no doubt that cohesin is enriched at loop anchors, but there is substantial disagreement among algorithms to call loops, so the "vast majority" may be a matter of preference for an algorithm over another.

14. Page 10, second sentence: "These can be separated into two main categories: loops that are CTCF dependent and CTCF independent, cell type specific, dynamic loops that form between enhancers and promoters and involve cell type specific TFs (52)". I do not understand this statement because I cannot see which are the two main categories.

15. Page 12, third paragraph, first sentence. I do not really understand the sentence "We also found preexisting CTCF independent contacts between H3K27ac enriched regions that fall within a loop that has enriched CTCF binding".

16. Page 13, third paragraph, third line: I do not really understand the sentence "Without any filtering we found that up or down regulation of the three variables was positively correlated in both TADs and CTCF-mediated loops as shown by the red and blue dots, where each dot represents a single TAD or CTCF-mediated loop (Supplementary Fig. 7a, b)".

17. Page 15, line 6: The small ovals in Figure 6b look like ovals only at high magnification. The authors could consider a different symbol to avoid confusion.

18. Page 15, line 6 from the end: "logistical regression" should be changed to the standard "logistic regression".

19. Page 16, second paragraph, line 3 from the end: "to drives" is a typo.

Reviewer #3 (Remarks to the Author):

In this manuscript, Lhoumaud et al. compare two isogenic human multiple myeloma cell lines that differ from the origin of the allele harboring disruption of the NSD2 gene. In the first cell line the mutated NSD2 allele is in its normal chromosomal environment, while in the second cell line the mutation is on the NSD2 allele that also bears a translocation with IgH. The resulting isogenic cell lines differ from the level of NSD2, providing an elegant, original and biologically relevant context in which to study the consequences of varying expression of this H3K36 methylase.

The various epigenomic experiments are very carefully conducted and analyzed, and the data stand of high quality. The manuscript is well written. I think this study is a very strong candidate for Nature Communications. Addressing the following points could strengthen the manuscript:

Major points

1. The expression levels of NSD2 between the two cell lines are not directly reported in the manuscript. Please include a figure displaying RNA levels, which can be directly derived from the RNA-seq data, as well as – if possible – western blot data to assess protein levels. Reporting bulk levels of H3K27me2 as well as H3K27me3 would also be important – western blot would be acceptable.

2. It is sometime difficult to understand what authors mean by change in compartmentalization. A given genomic bin can change its assignment to A or B, because of changes in the first Eigen vector of the PCA. Alternatively, compartmentalization strength may be overall changing, meaning how different are bins assigned to A compared to bins assigned to B. Absolute values of the first Eigen vectors are not always comparable because they are typically normalized during the PC decomposition. It would be helpful if authors could report the overall strength of compartmentalization using their Hi-C data, simply by performing a quantitative analysis of compartment strength as previously reported e.g. by Schwarzer et al. Nature (figure Supp 2d), Nora et al. Cell 2017 (figure Supp 4a) or Wutz et al. EMBO Journal 2017 (figure 1H).

3. Similarly, it would be helpful to provide a scatter plot of the Eigen vector 1 (either calculated in cis or in trans) for each genomic bin in the high cell line on X and low cell line on Y. This will indicate right away how many bins are different and how different they are.

4. I had a hard time understanding how many boundaries are affected and how much in the current figures. A simple scatter plotting the insulation score at each boundary with one cell line in X and the other in Y would be very helpful.

Minor points

5. It would be helpful to indicate in the short sentence in the introduction how the NSD2 alleles were inactivated, so readers do not need to pull out the original reference to understand the biological system.

6. Authors identify 260 and 278 super-enhancers in low and high cells. What fraction overlaps? A venn diagram in the main figure would be helpful

7. When saying "CTCF peaks [...] were significantly correlated with gene expression changes", It would be helpful if authors could refine their analysis of differential CTCF binding together with differential gene expression to display separately genes that are up- vs. down- regulated. This is important given that CTCF depletion was shown, as authors point out, to cause predominantly down-regulation of genes that are bound at promoters, not up-regulation.

8. From the figures, it is striking that the changes in H3K36me2 occur in a domain-wide fashion, with the boundaries of chromatin domains coinciding – at least in the examples shown – with TAD boundaries. It would be interesting if authors could expand on this, for example by identifying the boundaries of the domains that change homogenously in H3K36me2 and intersecting them with TAD boundaries. Alternatively, one could aggregate the analysis of H3K36me2 changes at the TAD boundaries in a meta-plot to test if TADs indeed tend to demarcate ruptures of H3K36me2 pattern changes.

9. Is the CTCF motif found in the new CTCF peaks gained in high NSD2 cells? If so, is does it deviate from the consensus? Could this explain why those sites may need a favorable chromatin state to be bound by CTCF?

We would like to thank the reviewers for taking the time to read our manuscript and provide us with helpful and constructive feedback. We have addressed all comments as outlined below.

Reviewers' comments:

Reviewer #1 (Remarks to the Author):

An interesting and comprehensive paper studying the epigenomic changes including the changes in enhancers, CTCF binding and chromatin looping that occur in response to high levels of MMSET/NSD2 in an isogenic pair of multiple myeloma cell lines. Major conclusions are that the changes in global chromatin modifications affects CTCF binding, the architecture of superenhancers and looping. This is an appropriate publication for Nature Communications after addressing some modest questions

1. Figure 2e H3K27Ac peaks are ranked, H3K36me2 levels were higher in these peaks but this may be due to the global increase in H3K36me2. What is the H3K36 levels of a randomly selected set of sites- The statement that the increased H3K27Ac is associated with H3K36me2 may be true/true and unrelated.

We have now included heatmaps and average profiles for ChIP-seq data at 1000 random sites (**Supplementary Fig. 4e-f**). The analysis reveals that although there is enrichment of H3K36me2 at random sites in NSD2 High cells this is not linked to an increase of H3K27ac, CTCF or ATAC-seq peaks. We now refer to these new panels in the results section page 10 as follows: "Although randomly selected sites show an enrichment of H3K36me2, this is not sufficient to drive chromatin accessibility and CTCF or H3K27ac enrichment (**Supplementary Fig. 4e-f**). This data suggests that spreading of H3K36me2 provides a more accessible chromatin landscape favorable for CTCF and H3K27ac enrichment at a subset of loci".

2. Figure 2e there is some decrease in H3K27me3 in the most noticeable H3K27Ac peaks, these two marks are mutually exclusive. Text on page 9 should be modified accordingly – I don't think that the increase in H3K27Ac is unrelated to H3K27me3.

We modified the text as suggested on page 9: "A slight decrease of H3K27me3 was observed at these locations (**Fig. 2d**) suggesting that increases in H3K27ac peaks may be partially dependent on H3K27me3 changes."

3. Again there is an increase in ATAC seq peaks associated with increased H3K36me2 – but this may be true at any site selected. Nevertheless it is clear that there are genes activated and more enhancers and more CTCF binding in the NSD2 high cells.

As per point 1 above, we have added a heatmap and average profile showing ATAC-seq peaks at 1000 random sites as requested by the reviewer (**Supplementary Fig. 4e-f**).

4. Figure 2f needs better description what do the red and blue track mean – I am not certain which is in the high versus low cells – figure 3 indicates that these may be subtraction tracks- please clarify this in the figure legend of figure 2. Some of the supplemental figures explicitly say that subtraction tracks or presented- please be consistent.

We have clarified this in the revised manuscript page 22 as follows: H-L refers to subtraction of the ChIP-seq signal (NSD2 High - Low). We have also checked all other figures for consistency.

The rest of the paper is really a comprehensive and extensive analysis of TADs compartments (please for readers define earlier on what are A and B compartments- this is defined in the supplemental but never in the main text) showing that oncogene up regulation is associated with switching to the active compartment, new CTCF and H3K27ac sites and stronger intra-TAD interactions. It is interesting that AP1 sites are enhanced in the H3K27Ac peaks, are there other motifs enriched as well?

We did in fact define this in the introduction on page 3: ‘Chromosomes are divided into large domains that physically separate into two nuclear compartments, A and B, of active and inactive chromatin, respectively ¹. We have also redefined this in the result section when we perform the analysis (page 11): “Active A and Inactive B compartment status was analyzed in merged NSD2 Low and High replicates using the Eigen vector (principal component 1, PC1, see Supplementary Information for details)”.

All the motifs enriched in increased and decreased H3K27ac peaks are shown in **Supplementary Fig. 2c** and referred to on page 8. We have now highlighted other examples in the discussion on page 18: “Indeed, our studies reveal that the motif for transcription factors such as FOXA1, CEBPA, AP-1 are enriched at increased H3K27ac peaks, including those designated as super-enhancers (**Supplementary Fig. 2c,d**).”

Reviewer #2 (Remarks to the Author):

Transparency statement: this review was written by Guillaume Filion (CRG, Barcelona). I accepted the assignment on May 17, 2019 and I turned in my review on May 29, 2019. I worked approximately 10 hours and 30 minutes on this task. I feel competent to review the genomics methods and the statistics but I lack the biological expertise in multiple myeloma. I have reviewed the GEO records and the statistics in the manuscript, but I have not replicated them.

In the article titled “NSD2 overexpression drives clustered chromatin and transcriptional changes in a subset of insulated domains”, Lhoumaud et al. investigate the dosage of the histone-methyltransferase NSD2 in a cell line model for multiple myeloma. The question is relevant because NSD2 can be up-regulated in multiple myeloma following a common translocation. Knocking out the two NSD2 alleles independently allows the authors to change the dosage of NSD2 in an isogenic context, and to observe the consequences directly.

The authors focus on gene expression, on the epigenome and on genome conformation. They observe changes at all levels, and find that they are usually consistent with the spread of the mark H3K36me2 (deposited by NSD2). For instance, H3K27ac and CTCF are typically gained in regions of heightened H3K36me2; with concomitant appearance of chromatin loops and changes in TAD borders and AB compartments.

Overall, the study is well conducted and the results are very coherent. This work is particularly relevant to tackle the question of the relationships between the epigenome and genome conformation, with a clinical relevance for cancer. In other words, I am very positive on the scientific quality of the work.

That said, I have some comments for improvements that mostly have to do with the statistics and the clarity of the text. The following issues are not all severe, but they are all important and I recommend the authors to address them the best they can.

1. There is too little information about the experimental design.

1a. Page 6, second sentence of the second paragraph: “Importantly, NTKO and TKO cells differ solely in the level of NSD2 they express and henceforth are referred to as NSD2 High and Low cells”. This is absolutely crucial for the study, yet the manuscript is not explicit whether the translocation leaves the gene NSD2 completely intact. This needs to be clarified in the text and possibly in the figures.

This is a good suggestion and we have now included the following statement in the revised version of the manuscript on page 6: “In order to address this question, we used two previously characterized isogenic cell lines generated from the patient-derived KMS11 t(4;14) multiple myeloma cell line: NTKO (non-translocated knockout) and TKO (translocated knockout) cells, which have the translocated allele or the endogenous *NSD2* allele, respectively inactivated by insertion of a stop codon just after exon 6, leading to a truncated protein that lacks all functional domains (Fig. 1a)². Importantly, NTKO and TKO cells genetically differ solely in the level of NSD2 they express and henceforth are referred to as NSD2 High and Low cells.”

1b. The GEO record refers to NSD2 High as wild type. This adds to the confusion and should be clarified on the record.

Thank you for pointing this out. The record has been appropriately altered. In addition, we have included a **Supplementary Table 4** to indicate the name of each sample, the cell line and which experiment they were used in.

1c. What is meant by “NSD2 High” and “NSD2 Low” exactly? Figure 1a labels normal plasma cells and “TKO” cells as “NSD2 Low”, and the KMS11 and “NTKO” cells as “NSD2 High”. For instance, was the parental KMS11 used for any experiment presented in the manuscript? The supplementary material for ATAC-seq mentions something about KMS11, but later specifies that only TKO and NTKO cells were used.

The authors seem to have two “TKO” clones, so how were replicates handled? Were the two clones systematically used in every experiment as “biological” replicates? For instance, in Supplementary Figure 1a, there are 4 points labelled NSD2 Low and 4 points labelled NSD2 High. Yet the legend only says “PCA of RNAseq replicates for NSD2 High and Low cells”. Does that mean two “technical” replicates of each NSD2 Low clone and four “technical” replicates of the NTKO cell? Meanwhile the RNA-seq paragraph mentions “RNA was extracted from 5 replicates for each condition”. If this is so, why are there only 4 points on Figure 1a? A little further, Supplementary Figure 2a shows 3 points each and the legend simply says “PCA of H3K27ac CHIP-seq replicates for NSD2 High and Low cells”. But now it is clear that there must be some imbalance in the design because there is an odd number of points.

Given the confusion about the meaning of “replicate” throughout the manuscript, I recommend the authors to update the method section to leave no doubt about this, nor about how replicates were discarded. For instance, the description of the ATAC-seq procedure is very clear (except the mention of KMS11 at the beginning), so the authors could just use this as a template for the other sections.

Thank you for this suggestion. “NSD2 High” refers to both the parental cell line KMS11 and the NTKO cell line while “NSD2 Low” refers to the TKO cell line. They were all used in the manuscript, and as mentioned above we have now added a **Supplementary Table 4**, where we indicate the name of each sample, the cell line and which experiment they were used in. For ATAC-seq, the KMS11 mention at the beginning was a mistake and this has been corrected in the revised manuscript.

For the RNA-seq replicates in the PCA, there are actually 5 dots per condition which are not all clearly visible as some replicates strongly overlap. The 5 replicates are more visible in **Supplementary Fig. 1b** displaying normalized reads counts for each of them. We have now used the description of the ATAC-seq as a template and added further details about replicates in the supplementary information under each section in the method section of the revised supplementary information file.

2. I would like to raise the following issues with the statistics.

2a. The criteria for significance keep changing throughout the manuscript. Sometimes they are based on p-values, sometimes they are based on FDR. For p-values, the cut-off for significance seems to be 0.05, but for FDR it can be 0.1, 0.05, or 0.01. I strongly recommend the authors to uniformize their approach to prevent unconscious p-hacking. At the very least they have to explain why they switch from FDR to p-values and back, and what is the motivation for using a given cut-off. My personal opinion (which I do not wish to impose) is to use FDR with a fixed cut-off for selection procedures and p-values for single tests. For instance I would have used FDR to represent the data of Figure 1g.

For each p-value or FDR mentioned, we have now indicated the tool used for each statistical test, the cutoff applied and whether default parameters were used. This information has now been included in **Supplementary Table 5** referred to as follow in the Supplementary Information, page 17: "Statistical analysis: **Supplementary Table 5** indicates the tool used for each statistical test, the cutoff applied and whether default parameters were used."

As per the reviewer's suggestion, p-value has been changed for FDR in **Fig. 1f** as well as **Supplementary Fig. 2c**.

For DEseq2 analysis, we changed the cutoff for 4C-seq (**Fig. 6b**) from 0.05 to 0.01 to be more consistent with other DEseq2 analysis. An exception was made for super-enhancers for which we kept FDR <0.1. Results from analysis with an FDR <0.01 are available for the reviewer are shown in the figure below. Only 30 super-enhancers are now called as significant (11 decreased and 19 increased in NSD2 High cells). The violin plot is very similar to the one displayed in **Fig. 1g** but TF motifs were less significant with the more stringent FDR. If the reviewer prefers we can replace the original analysis with a more stringent cutoff.

Other FDRs were determined based on default parameters of the tools used (see **Supplementary Table 5**).

a 30 differentially regulated super-enhancers (SEs) in NSD2 High versus Low cells

b SEs are associated with gene expression

c Motifs found in ATAC-seq peaks in differential SEs

2b. The reports are sometimes incomplete. For instance, the statistical tests are not always described when a p-value is provided.

As mentioned above we have a new Supplementary Table 5 which indicates the tool used for each statistical test, the cutoff applied and whether default parameters were used.

2c. The reports of the FDR are sometimes awkward. For instance, one can read "... FZD8 (log2 fold-change = 3.16, FDR = 8.18E-34)". The concept of FDR is by definition attached to a set and not to an individual (here a gene). Algorithms usually compute a certain FDR score in order to include or exclude an individual in the set, but unlike p-values, each individual FDR score depends on all the genes. Thus, considering the value in isolation is not so meaningful. It is more coherent to simply say that the gene is in a set of chosen FDR. This also avoids the problem of reporting "exotic values" such as 8.18E-34, which give a false impression of confidence.

We have followed the reviewer's suggestion.

3. Page 17, third paragraph, first sentence: "Since the NSD2 phenotype relies on the histone methyl transferase activity of the protein". I somehow missed the evidence where the authors prove this claim in their model. The rest of the paragraph is to the point, but here the authors should show the evidence that the methyltransferase activity and no other potential function of NSD2 drives the changes they observe (perhaps was it done by the creators of the cell line), or they should rephrase the conclusion.

This finding was made by two other groups, which used catalytic mutants of NSD2 to demonstrate that the NSD2 high phenotype relies on HMT activity. We now mention in the text that it was previously described in these reports and have included the references. Page 18: "Since the NSD2 phenotype relies on the histone methyl transferase activity of the protein, as previously described^{3,4}, the upstream causal event is the expansion of H3K36me2 domains."

4. As currently described, the logistic model recapitulates the previous results but adds little insight. The results of Figures 7a and 7b would be better presented as a statistical test (using Fisher's exact test for instance) because the model is used neither for inference nor for prediction. This is somewhat unfortunate because the chosen vocabulary suggests a causal relationship (e.g., "We found that changes in H3K27ac significantly increase the probability that a gene sharing the same CTCF loop is differentially expressed") whereas the discussion clearly shows that this is not the intention of the authors. A relatively easy way to improve this part is to use the model to test the interaction term between CTCF and H3K27ac in order to see whether they synergize to account for gene expression. This is typically what models are used for, so it would justify the approach. Alternatively, the authors could replace this part by a standard descriptive approach coupled to a test.

Fisher "exact" test is not appropriate for this scenario as the marginal of the contingency table is not fixed in advance. Generalized linear models, like logistic regression models, offer greater flexibility to model more complex scenarios, like the present one, in addition to de-coupling the effects of the 2 overlapping predictors. Moreover, most statistical tests are equivalent to some linear model and we believe that including more information like effect size as well as p-values increases the value of the analysis. The models are used for inference and we infer the effect that H3K27ac and CTCF differential peaks have on the probability of a randomly selected gene to be differentially expressed. We agree that the wording of the particular phrase may mislead the reader to believe that we imply there is a causal relationship and thus we re-worded it to "We found that differentially expressed genes are significantly enriched within the same CTCF loop as altered H3K27ac peaks. Furthermore, differentially expressed genes are significantly enriched within the same TAD as altered H3K27ac and CTCF peaks." on page 16, as well as in the abstract, the introduction page 5 and the discussion page 17. The model was used to test for interaction between the 2 terms but none of them came up as significant. For this reason we did not include them in the manuscript. We added the sentence "we tested for possible synergistic or

antagonistic effects of H3K27ac and CTCF peaks and neither was significant” on page 16.

The following issues are less important scientifically, but they damage the clarity of the manuscript, so I recommend the authors to address them (most of them are fairly easy to fix).

1. Abstract: “a linear regression model” is referred to as a logistic regression model in the text. I believe the latter is the correct one, but in any case the text has to be consistent.

Thank you identifying this error. We have made the change from “linear regression model” to “logistic regression model”.

2. First paragraph, page 3: “1MB sized highly self-interacting ‘topologically associated domains’ (TADs)” is inaccurate. TADs are not all 1MB in size, so “MB sized” or “Mb size” or “Mbp sized” would be more appropriate. Also, “highly self-interacting” is subjective. Actually, polymer models suggest the opposite. For instance, the work by Luca Giorgetti et al. on the X chromosome (PMID: 24813616, figure 2G) shows that many structures are compatible with the same TAD, many of which are not particularly self interacting. Some unpublished work by Geoffrey Fudenberg and Leonid Mirny also shows that TADs are nowhere near “isolated” as a polymer folded in 3D space. Here the authors should either stick to objective definitions or back up their statement with a citation so that we know what they mean by “highly self-interacting”.

We changed the sentence in the revised manuscript as suggested by the reviewer: Compartments can be further subdivided into Mbp sized ‘topologically associating domains’ (TADs).

3. Page 4, 7 and 8 lines from the end: the acronym HMT is defined but never used in the manuscript. The acronym MM is defined and used only in Figure 1a. The best option is to not define the acronym HMT at all (the text reads well without it) and to define the meaning of MM in the legend of Figure 1a. Also not that “MM cell lime” should be “MM cell line” in the panel.

We have removed the acronym HMT from page 4 and defined MM in the legend of Fig. 1a.

4. Page 5, 9 lines from the end: Rad21 appears for the first time in the manuscript, without mention that it a cohesin component. This will be mentioned at the next mention of Rad21 in the text, so the authors should move the precision “(a component of the cohesin complex)” here.

We have made the suggested change in the revised manuscript.

5. Page 6, first sentence of the second paragraph: it would be so much easier to understand if “a patient derived” were replaced by “the patient-derived”. This looks like a detail, but I actually had to run a Google search on KMS11 to make sure that I understood the sentence properly. The same comment applies to the legend of Figure 1a.

We have made the suggested changes in the revised manuscript.

6. Page 6, last sentence of the second paragraph: “Genes associated with multiple myeloma and KRAS pathways were enriched as shown by Gene Set Enrichment Analysis (GSEA, Supplementary Fig. 1b)”. This statement is not clear. In which context are those genes enriched, and relative to what? It surmise that this means that the 1953 genes that are differentially expressed contain more genes associated

with myeloma and KRAS than expected from a random set of 1953 genes, but this is not clear without more context information.

All genes were considered for GSEA to get the enriched pathways. We changed the sentence on page 6 to clarify: “MM and KRAS pathways were enriched (FDR < 0.25) as shown by GSEA (**Supplementary Fig. 1d**) based on the fold change of expression in NSD2 High vs Low cells across all genes from DEseq2 analysis (26586 genes, see Method for details)”. More information was added in the method section of Supplementary information: “Gene Set Enrichment Analysis for pathway enrichment was performed using the default parameters of the GSEA desktop application^{5,6} on normalized read counts for each NSD2 High and Low RNA-seq replicate from the 26586 protein coding genes of hg38 genome.”

7. Page 6, two lines from the end: “Since the vast majority of regulatory elements are located within intergenic regions”. Is this the case? Figure 1g suggests the opposite. Here the authors could backup their statement with a citation, and refine it for consistency with their own results because they seem to distinguish intergenic regions from promoters.

This sentence is changed as follows: “To determine whether spreading of H3K36me2 into intergenic regions could impact the activation status of potential regulatory elements, we analyzed H3K27ac, a feature of both enhancers and promoters”, (page 7).

8. What is the information of Figure 1d? It is mentioned in the text together with figure 1c, but no specific comment is made about it. Similarly, what is the information of Figure 2b? The authors should either comment these panels and explicitly mention why they are relevant for their scientific argument, or drop them altogether.

Volcano plots and heatmaps provide similar information and these have now been combined in one figure panel. We have clarified this as follows: **Fig. 1c** is now labelled: “2597 differentially regulated H3K27ac peaks in NSD2 High versus Low cells” and the figure legend has been altered as follows: “Violin plot (left panel) and heatmap (right panel) showing significant NSD2-mediated changes in H3K27ac (adjusted p-value <0.01, Wald test). Decreased peaks: 701 (blue, log2 fold change <-1); increased peaks: 1896 (red, log2 fold change >1).” Similarly, **Fig. 2a** is now labelled “2492 differential CTCF peaks in NSD2 High versus Low cells” and the figure legend has been changed to: “Violin plot (left panel) and heatmap (right panel) showing significant NSD2-mediated changes in CTCF binding (adjusted p-value <0.01, Wald test).”

9. Page 8, second paragraph, second sentence from the end: “possibly because this is the most accurate prediction”. I do not understand the meaning of this statement. The authors should clarify it or remove it.

We have removed this statement.

10. Page 8, 5 lines from the end: “precise transcriptional control is exerted by restricting the influence of enhancers to target genes within TADs whose boundaries are enriched for the insulating protein, CTCF”. This statement could be backed up by a citation so that we know what the authors have in mind here. For instance, this is dubious in *Drosophila* where TADs are not really flanked by CTCF sites. Likewise, transcriptional control is sometimes achieved in different ways. Think of how viruses can tightly regulate the expression of their genes independently of enhancers. In summary, giving some context would make this statement more concrete.

We have revised this statement as follows (page 8): “In particular, TADs (whose boundaries are enriched for the insulating protein CTCF) generate a 3D chromatin structure that has been proposed to delimit the action of enhancers to genes within the same domain ⁷⁻¹⁰.”

11. Page 9, line 4: “Increased CTCF peaks were predominantly located in intergenic and intragenic regions, as compared with decreased and stable CTCF peaks (Fig. 2c)”. I do not understand this statement. Perhaps “and intragenic” was not meant to be there?

We have replaced the word intragenic with intronic to clarify.

12. Page 9, second paragraph, second sentence: the logic of this passage is somewhat confusing. “Importantly, new H3K27ac and CTCF peaks were found to be located in newly enriched H3K36me2 regions while H3K27me3 remained unchanged at these locations”. The grammar of this sentence suggests that H3K27me3 does not change at the newly enriched H3K36me2 regions, but the figure shows that it does not change at the new H3K27ac and CTCF peaks. For this reason, the next sentence “Thus, increases in H3K27ac peaks were found to be independent of H3K27me3 changes” does not appear as the logical consequence of the previous one. Clarifying the first sentence would solve the issue.

As mentioned above, we changed the sentences in the revised manuscript as follows: “We observed enriched H3K36me2 levels at increased H3K27ac and CTCF peaks as opposed to stable and decreased H2K27ac and CTCF peaks (Fig. 2d and Supplementary Fig. 4). A slight decrease of H3K27me3 was observed at these locations (Fig. 2d) suggesting that increases in H3K27ac peaks may be partially dependent of H3K27me3 changes.”

13. Page 10, first sentence: “The vast majority of chromosomal loops involve cohesin”. This should be backed up by a citation. There is no doubt that cohesin is enriched at loop anchors, but there is substantial disagreement among algorithms to call loops, so the “vast majority” may be a matter of preference for an algorithm over another.

We have clarified as follows on page 10: “Cohesin’s importance in chromatin looping has been widely demonstrated ¹¹⁻¹⁵.”

14. Page 10, second sentence: “These can be separated into two main categories: loops that are CTCF dependent and CTCF independent, cell type specific, dynamic loops that form between enhancers and promoters and involve cell type specific TFs (52)”. I do not understand this statement because I cannot see which are the two main categories.

We now have clarified as follow on page 10: “Cohesin’s importance in chromatin looping has been widely demonstrated ¹¹⁻¹⁵. Loops can be separated into CTCF dependent and CTCF independent categories. Specifically, CTCF independent cohesin-mediated loops are cell type specific, dynamic loops that form between enhancers and promoters and involve cell type specific TFs ¹⁶.”

15. Page 12, third paragraph, first sentence. I do not really understand the sentence “We also found preexisting CTCF independent contacts between H3K27ac enriched regions that fall within a loop that has enriched CTCF binding”.

We have clarified as follows (page 13): “We also found contacts between H3K27ac enriched regions within domains bound by enriched CTCF.”

16. Page 13, third paragraph, third line: I do not really understand the sentence “Without any filtering we found that up or down regulation of the three variables was positively correlated in both TADs and CTCF-mediated loops as shown by the red and blue dots, where each dot represents a single TAD or CTCF-mediated loop (Supplementary Fig. 7a, b)”.

We have clarified as follows (page 14): “The three variables are positively correlated in both TADs and CTCF-mediated loops, without considering any fold change or FDR thresholds. (Supplementary Fig. 9a, b).”

17. Page 15, line 6: The small ovals in Figure 6b look like ovals only at high magnification. The authors could consider a different symbol to avoid confusion.

We have replaced the figure with one that has bigger circles.

18. Page 15, line 6 from the end: “logistical regression” should be changed to the standard “logistic regression”.

We have corrected the error.

19. Page 16, second paragraph, line 3 from the end: “to drives” is a typo.

We have corrected the error.

Reviewer #3 (Remarks to the Author):

In this manuscript, Lhoumaud et al. compare two isogenic human multiple myeloma cell lines that differ from the origin of the allele harboring disruption of the NSD2 gene. In the first cell line the mutated NSD2 allele is in its normal chromosomal environment, while in the second cell line the mutation is on the NSD2 allele that also bears a translocation with IgH. The resulting isogenic cell lines differ from the level of NSD2, providing an elegant, original and biologically relevant context in which to study the consequences of varying expression of this H3K36 methylase.

The various epigenomic experiments are very carefully conducted and analyzed, and the data stand of high quality. The manuscript is well written. I think this study is a very strong candidate for Nature Communications. Addressing the following points could strengthen the manuscript:

Major points

1. The expression levels of NSD2 between the two cell lines are not directly reported in the manuscript. Please include a figure displaying RNA levels, which can be directly derived from the RNA-seq data, as well as – if possible – western blot data to assess protein levels. Reporting bulk levels of H3K27me2 as well as H3K27me3 would also be important – western blot would be acceptable.

We have now added RNAseq counts as well as western blots showing differences in the levels of NSD2, H3K36me2 and H3K27me3 in the NSD2 High versus Low cells (Supplementary Fig. 1b and c).

2. It is sometime difficult to understand what authors mean by change in compartmentalization. A given genomic bin can change its assignment to A or B, because of changes in the first Eigen vector of the PCA. Alternatively, compartmentalization strength may be overall changing, meaning how different are bins assigned to A compared to bins assigned to B. Absolute values of the first Eigen vectors are not always comparable because they are typically normalized during the PC decomposition. It would be helpful if authors could report the overall strength of compartmentalization using their Hi-C data, simply by performing a quantitative analysis of compartment strength as previously reported e.g. by Schwarzer et al. Nature (figure Supp 2d), Nora et al. Cell 2017 (figure Supp 4a) or Wutz et al. EMBO Journal 2017 (figure 1H).

We have now added compartment strength plots and values in **Supplementary Fig. 5c**, showing an overall decrease in compartment strength in NSD2 High as compared to Low cells using compartment strength calculations from both Schwarzer et al. Nature 2017 and Nora et al. Cell 2017^{14,17}.

Of note, we would like to clarify that in our compartment analysis we compared the interaction profiles between the conditions to define the regions with compartment changes (we only compared eigenvalues to visualize a trend in the plots in **Fig. 3a**). We used the HOMER pipeline (getHiCcorrDiff.pl) to correlate the interaction profile of each locus in one condition to the interaction profile of that same locus in another condition as explained in the HOMER Software website. We mention this in the Method section (page 21 in the supplementary file): "Using Homer's getHiCcorrDiff.pl, we compared the interaction profiles of both conditions and obtained a correlation difference to identify stable and switching compartments".

3. Similarly, it would be helpful to provide a scatter plot of the Eigen vector 1 (either calculated in cis or in trans) for each genomic bin in the high cell line on X and low cell line on Y. This will indicate right away how many bins are different and how different they are.

A density plot of the Eigen vector 1 is now included in **Supplementary Fig. 5d**.

4. I had a hard time understanding how many boundaries are affected and how much in the current figures. A simple scatter plotting the insulation score at each boundary with one cell line in X and the other in Y would be very helpful.

Scatter plots of boundary scores are now included in **Supplementary Fig. 6c**.

Minor points

5. It would be helpful to indicate in the short sentence in the introduction how the NSD2 alleles were inactivated, so readers do not need to pull out the original reference to understand the biological system.

We have added the following information on page 6: "In order to address this question, we used two previously characterized isogenic cell lines generated from the patient-derived KMS11 t(4;14) multiple myeloma cell line: NTKO (non-translocated knockout) and TKO (translocated knockout) cells, which have the translocated allele or the endogenous NSD2 allele, respectively inactivated by insertion of a stop codon just after exon 6, leading to a truncated protein that lacks all functional domains (**Fig. 1a**)²."

6. Authors identify 260 and 278 super-enhancers in low and high cells. What fraction overlaps? A venn diagram in the main figure would be helpful.

We have now included a Venn diagram in **Supplementary Fig. 2b** to clarify and refer to this in the text on page 7 as follows: “With this approach, we identified 260 and 278 super-enhancers in NSD2 low and high cells, respectively, with 110 overlapping between conditions^{18,19} (**Supplementary Fig. 2b**).” Because the approach illustrated by the Venn diagram was not used for the analysis, we thought it would be more appropriate to add the Venn diagram in the Supplementary figure together with the ROSE cutoffs. Alternatively it could be moved to the main figure if the reviewer thinks that it is more relevant.

7. When saying “CTCF peaks [...] were significantly correlated with gene expression changes”, It would be helpful if authors could refine their analysis of differential CTCF binding together with differential gene expression to display separately genes that are up- vs. down- regulated. This is important given that CTCF depletion was shown, as authors point out, to cause predominantly down-regulation of genes that are bound at promoters, not up-regulation.

We have intersected genes associated with CTCF peaks with up and downregulated genes, and indicate the number in a table in **Fig. 2c** on top of the violin plots. This table shows that a large portion of differentially expressed genes are associated with stable CTCF peaks. Reference to this table was added to the text on page 9: “Moreover, although a large portion of differentially expressed genes are associated with stable CTCF peaks, CTCF peaks that were enriched or depleted at promoters and distal sites were significantly correlated with gene expression changes, with the strongest effect seen at promoters (**Fig. 2c**).” The figure legend was changed as follows on page 22: “Gene expression changes are associated with CTCF changes (top panel, 0-50kb up and downstream from TSSs associated with genes using GREAT, upreg. and downreg. refers to upregulated and downregulated genes), at promoters (top middle panel, +/- 3kb around TSS as defined using ChIPseeker in **Fig. 2b**) and distal CTCF sites (bottom panel, 3-250kb up and downstream from TSSs associated with genes using GREAT). CTCF up, down and stable are shown respectively as red, blue and grey in NSD2 High versus Low cells.”

8. From the figures, it is striking that the changes in H3K36me2 occur in a domain-wide fashion, with the boundaries of chromatin domains coinciding – at least in the examples shown – with TAD boundaries. It would be interesting if authors could expand on this, for example by identifying the boundaries of the domains that change homogeneously in H3K36me2 and intersecting them with TAD boundaries. Alternatively, one could aggregate the analysis of H3K36me2 changes at the TAD boundaries in a meta-plot to test if TADs indeed tend to demarcate ruptures of H3K36me2 pattern changes.

Average profiles of H3K36me2 in NSD2 High versus Low cells at changing boundaries (61 boundaries shown in **Fig. 4a**) as compared to 61 random stable boundaries have now been included in **Supplementary Fig. 6d**. This plot shows that on average H3K36me2 is higher in NSD2 High versus Low cells at both boundaries and surrounding regions, for the changing and stable boundaries. This is commented in the text on page 12 as: “H3K36me2 levels are higher in NSD2 High versus Low cells both at boundaries and surrounding regions. However, we could not detect any difference in H3K36me2 enrichment in the NSD2 high cells at boundaries with increased strength versus stable boundaries (**Supplementary Fig. 6d**).”

Of note, a similar analysis was performed at switching compartment regions (**Supplementary Fig. 5h**) showing an increase of H3K36me2 at regions that switch from B to A, compared to regions that switch from A to B (consistent with **Fig. 3e**). However, the average profiles in **Supplementary Fig. 5h** show that an increase of H3K36me2 is not limited to switching regions as it expands outside of these regions. We refer to these new plots on page 11 as: “Compartment switching from B to A overlapped with increased

H3K36me2 levels and a reduction in H3K27me3, while the reverse was true for switching from compartment A to B (**Fig. 3d, e and Supplementary Fig. 5h**). Of note, an increase in H3K36me2 is not limited to switching regions (**Supplementary Fig. 5h**)."

9. Is the CTCF motif found in the new CTCF peaks gained in high NSD2 cells? If so, is does it deviate from the consensus? Could this explain why those sites may need a favorable chromatin state to be bound by CTCF?

We used MEME-ChIP to look at CTCF motifs at the 2197 peaks with increased signal in NSD2 High versus Low cells. As a control, we randomly selected 2197 CTCF peaks from the full list of peaks. CTCF motifs from the MEME-ChIP are shown in **Supplementary Fig. 3b** together with MA0139.1 from the Jaspas database as a reference. Overall, the motifs look similar. We indicate this in the results section on page 9 as follows: "Increased CTCF peaks harbor a similar motif to that observed for randomly selected CTCF peaks and the consensus motif MA0139.1 from the Jaspas database (**Supplementary Fig. 3b**)." The numbers of CTCF sites and the E-value from MEME ChIP are indicated in the figure.

The figure legend has been altered as follows: "b, CTCF motifs in CTCF ChIP-seq peaks identified using MEME-ChIP tool from the MEME suite^{20,21} (2197 increased peaks from **Fig. 2a**, (top panel), 2197 randomly selected CTCF peaks from the full list of peaks, (middle panel) and the CTCF motif MA0139.1 from the Jaspas database (bottom panel). The numbers of CTCF sites and the E-value from MEME-ChIP are indicated." We also added the following in the method section under "ChIP-seq Data processing and quality control": "For CTCF ChIP-seq, the MEME-ChIP tool from the MEME suite^{20,21} was used to detect CTCF motifs in increased peaks and equivalent number of peaks randomly picked from the full list of peaks."

1. Lieberman-Aiden, E. *et al.* Comprehensive mapping of long-range interactions reveals folding principles of the human genome. *Science* **326**, 289-93 (2009).
2. Lauring, J. *et al.* The multiple myeloma associated MMSET gene contributes to cellular adhesion, clonogenic growth, and tumorigenicity. *Blood* **111**, 856-64 (2008).
3. Kuo, A.J. *et al.* NSD2 links dimethylation of histone H3 at lysine 36 to oncogenic programming. *Mol Cell* **44**, 609-20 (2011).
4. Popovic, R. *et al.* Histone methyltransferase MMSET/NSD2 alters EZH2 binding and reprograms the myeloma epigenome through global and focal changes in H3K36 and H3K27 methylation. *PLoS Genet* **10**, e1004566 (2014).
5. Subramanian, A. *et al.* Gene set enrichment analysis: a knowledge-based approach for interpreting genome-wide expression profiles. *Proc Natl Acad Sci U S A* **102**, 15545-50 (2005).
6. Mootha, V.K. *et al.* PGC-1alpha-responsive genes involved in oxidative phosphorylation are coordinately downregulated in human diabetes. *Nat Genet* **34**, 267-73 (2003).
7. Andrey, G. & Mundlos, S. The three-dimensional genome: regulating gene expression during pluripotency and development. *Development* **144**, 3646-3658 (2017).
8. de Laat, W. & Duboule, D. Topology of mammalian developmental enhancers and their regulatory landscapes. *Nature* **502**, 499-506 (2013).
9. Franke, M. *et al.* Formation of new chromatin domains determines pathogenicity of genomic duplications. *Nature* **538**, 265-269 (2016).
10. Symmons, O. *et al.* Functional and topological characteristics of mammalian regulatory domains. *Genome Res* **24**, 390-400 (2014).

11. Fudenberg, G. *et al.* Formation of Chromosomal Domains by Loop Extrusion. *Cell Rep* **15**, 2038-49 (2016).
12. Hansen, A.S., Pustova, I., Cattoglio, C., Tjian, R. & Darzacq, X. CTCF and cohesin regulate chromatin loop stability with distinct dynamics. *Elife* **6**(2017).
13. Rao, S.S.P. *et al.* Cohesin Loss Eliminates All Loop Domains. *Cell* **171**, 305-320 e24 (2017).
14. Schwarzer, W. *et al.* Two independent modes of chromatin organization revealed by cohesin removal. *Nature* **551**, 51-56 (2017).
15. Wendt, K.S. *et al.* Cohesin mediates transcriptional insulation by CCCTC-binding factor. *Nature* **451**, 796-801 (2008).
16. Snetkova, V. & Skok, J.A. Enhancer talk. *Epigenomics* **10**, 483-498 (2018).
17. Nora, E.P. *et al.* Targeted Degradation of CTCF Decouples Local Insulation of Chromosome Domains from Genomic Compartmentalization. *Cell* **169**, 930-944 e22 (2017).
18. Hnisz, D. *et al.* Super-enhancers in the control of cell identity and disease. *Cell* **155**, 934-47 (2013).
19. Lorenz, J. *et al.* From CLL to Multiple Myeloma - Spleen Tyrosine Kinase (SYK) influences multiple myeloma cell survival and migration. *Br J Haematol* **174**, 985-9 (2016).
20. Bailey, T.L. *et al.* MEME SUITE: tools for motif discovery and searching. *Nucleic Acids Res* **37**, W202-8 (2009).
21. Beard, C., Hochedlinger, K., Plath, K., Wutz, A. & Jaenisch, R. Efficient method to generate single-copy transgenic mice by site-specific integration in embryonic stem cells. *Genesis* **44**, 23-8 (2006).

REVIEWERS' COMMENTS:

Reviewer #1 (Remarks to the Author):

My modest suggestions were all answered in a completely satisfactory manner by additional data analysis and modifications of the text

Reviewer #2 (Remarks to the Author):

The authors have addressed all my concerns.

Reviewer #3 (Remarks to the Author):

Authors have addressed all my suggestions and I recommend publication without delay.